# Complete mitochondrial genome of *Episymploce splendens* (Blattodea: Ectobiidae): A large intergenic spacer and lacking of two tRNA genes

**Lin Yan**[1], **Zhenzhen Hou**[2], **Jinnan Ma**[1], **Hongmei Wang**[1], **Jie Gao**[3], **Chenjuan Zeng**[3], **Qin Chen**[1], **Bisong Yue**[1], **Xiuyue Zhang**[1,2]*

**1** Key Laboratory of Bio-Resources and Eco-Environment, Ministry of Education, College of Life Sciences, Sichuan University, Chengdu, China, **2** Sichuan Key Laboratory of Conservation Biology on Endangered Wildlife, College of Life Sciences, Sichuan University, Chengdu, China, **3** Sichuan Key Laboratory of Medicinal Periplaneta Americana, Sichuan Gooddoctor Pharmaceutical Group, Chengdu, China

* zhangxiuyue@scu.edu.cn

**Data Availability Statement:** Our Data has been uploaded to NCBI Genbank database and is available at the GenBank Accession number

## Abstract

The complete mitochondrial genome of *Episymploce splendens*, 15,802 bp in length, was determined and annotated in this study. The mito-genome included 13 PCGs, 20 tRNAs and 2 rRNAs. Unlike most typical mito-genomes with conservative gene arrangement and exceptional economic organization, *E. splendens* mito-genome has two tRNAs (tRNA-Gln and tRNA-Met) absence and a long intergenic spacer sequence (93 bp) between tRNA-Val and srRNA, showing the diversified features of insect mito-genomes. This is the first report of the tRNAs deletion in blattarian mito-genomes and we supported the duplication/random loss model as the origin mechanism of the long intergenic spacer. Two Numts, Numt-1 (557 bp) and Numt-2 (975 bp) transferred to the nucleus at about 14.15 Ma to 22.34 Ma, and 19.19 Ma to 24.06 Ma respectively, were found in *E. splendens*. They can be used as molecular fossils in insect phylogenetic relationship inference. Our study provided useful data for further studies on the evolution of insect mito-genome.

## Introduction

The mitochondrial genome has the characteristics of maternal inheritance, rare genetic recombination, faster evolution and conservative gene arrangement [1]. The mito-genome commonly displays exceptional economy of organization, with overlapping genes or small intergenic spacers [2]. A typical insect mito-genome, with a compact circular molecule, generally encodes a fixed set of 37 genes including 13 protein-coding genes (PCGs), 22 transfer RNA (tRNA) genes, two ribosomal RNA (rRNA) genes and a control region [3]. However, exceptions have been reported, such as long gene intergenic spacer [4–6], gene rearrangement and gene loss [7, 8]. So far, mitochondrial gene rearrangement has been reported in about 17 orders of insects, and these genes include protein coding gene, rRNA gene and tRNA gene [8]. In addition tRNAs, the missing protein coding genes have also been reported in Mantodea,

OK094023 (https://www.ncbi.nlm.nih.gov/search/all/?term=OK094023).

**Funding:** This study was funded by the Special funds for central government to guide local scientific and Technological Development (2020ZYD098) and National Natural Science Foundation of China (U21A20409).The funders had no role in study design, data collection and analysis, decision to publish, or preparation of the manuscript.

**Competing interests:** The authors have declared that no competing interests exist.

Phthiraptera, and Psocoptera [9–13]. The changes of tRNAs are most various in no-typical insect mito-genomes, including tRNA translocation, tRNA loss, tRNA tandem duplication and tRNA conversion [4, 9–10, 14, 15]. Phthiraptera and Psocoptera insect mito-genomes may be most multipartite and most fast in evolution, that they could fission into fragmented mito-genomes and have numerous pseudo-genes and diverse gene rearrangements [7–10]. Thus, insect mitochondrial genomes are good models for researching mitochondrial genome evolution.

There are of 4,600 species of cockroaches [16] and 300 mitochondrial genomes of them have been sequenced and uploaded to the NCBI database. The blattarian mito-genomes seem to be conserved in evolution [16, 17]. The gene rearrangement was only reported in Cryptocercidae within blattarian mito-genomes [4]. *Episymploce splendens* belongs to Ectobiidae. Previous studies have mainly focused on the morphological features of *Episymploce* rather than molecular data [18, 19]. At present, no complete mitochondrial sequences of *Episymploce* have been recorded in the NCBI database. To obtain the sequence information and organization features of mito-genomes in *Episymploce*, we sequenced, annotated and described the complete mito-genome of *E. splendens*. Two tRNAs loss and a long intergenic region was found in *E. splendens* mito-genome and two pseudo-genes were also identified in the study. This study could deepen our understanding of mitochondrial genome of insects and contribute to the study of insect mito-genome evolution.

## Materials and methods

### Sample and DNA extraction

The cockroaches used in this study were collected on Mount Emei, Sichuan Province, China. The fresh material was placed in absolute ethanol and stored at -20°C. Total genomic DNA was extracted from the muscle tissue (legs) using TIANamp Genomic DNA kit (TIANGEN, Beijing, China). The concentration and purity of total DNA were detected by spectrophotometer. In addition, DNA was detected by agarose gel electrophoresis, and 1% agarose gel electrophoresis judged whether DNA was successfully extracted or not. Finally, DNA was stored at -20°C.

### PCR amplification and sequencing

Primers were designed based on the conserved sequences of *Blattella germanica* and *Blattella bisignata*, and then specific primers were designed based on the amplified and sequenced sequences at both ends [20, 21]. The software Primer Premier 5.0 was used to designed primers and the primer details are listed in Table 1. Primers Es3, Es9 and Es10 were obtained from Xiao *et al* [10]. PCR was conducted using 2×Taq PCR Mix (Innovagene, Chengdu, China) and performed on a PTC-100 thermal cycler (BioRad, Hercules, CA) with the following cycling conditions: an initial denaturation for 5 min at 94°C, followed by 35 cycles of denaturation for 30s at 94°C, annealing for 30s at 50–62°C (depending on primer combinations), elongation for 1–4 min (depending on putative length of the fragments) at 72°C, and a final extension step of 72°C for 10 min. PCR products were estimated by 1.0% agarose gel electrophoresis and sent to Tsingke Biotechnology Company (Chengdu, China) for sequencing. All fragments were bidirectional sequences, and the unsuccessful sequenced fragments were redesigned with primers and sequenced again to complete the sequence.

The sequences which were transferred to nuclear DNA from mitochondrial genome were named Numts [22]. Primers that amplified mitochondrial genes may occasionally amplify Numts [23]. In this study, agarose gel electrophoresis of the amplification product (the primer Numt-1-1 and Numt-2-1, Table 1) appeared as two bands. We then redesigned primers

**Table 1. Primers for the PCR amplifications of *Episymploce splendens* (Es) mito-genome and Numts.**

| Primer name | Upstream primers sequences(5'-3') | Downstream primers sequences(5'-3') | Anneal temperature (˚C) | Extension time (Second) |
|---|---|---|---|---|
| Es1 | 140-CCTCTCTTATCGCAATGTCCA | 1609-CGTGGGAAAGCTATATCAGGA | 58 | 90 |
| Es2 | 1367-GGTCAACAAATCATAAAGATATTGG | 2074-TAAACTTCAGGGTGACCAAAAAATCA | 55 | 45 |
| Es3 | 1559-CAACATTTATTTTGATTCTTTGG | 3649-GTTTAAGAGACCACCACTTG | 50 | 90 |
| Es4 | 3401-GAAGACTTTCACCAACCATC | 4391-TAGTACACTCATCTACTCTGGTAAC | 52 | 60 |
| Es5 | 4031-TGTAACAGCCCATGCTT | 5797-AATCGCAATGATGGTAGG | 51 | 110 |
| Es6 | 4945-AGAAGACTTTCACCAACCAT | 6997-GGATTCTCAAGATATTCGTT | 51 | 120 |
| Es7 | 6378-TCAACCGTTATCGAAAGACT | 7328-CTCCTACTCCTGTATCTGCTT | 52 | 60 |
| Es8 | 7010-AACGAATATCCTGAGAATCC | 8415-CACGGATTATGTTCTTCAGG | 52 | 90 |
| Es9 | 8290-GAAGGGGGTGCTGCTATATTAC | 11151-ATTACTCCTCCTAATTTATTAGGAAT | 62 | 180 |
| Es10 | 10491-CAATGAGTATGAGGAGGATTTGCTGT | 14755-TGTGCCAGCAGTCGCGGTTATACA | 59.5 | 240 |
| Es11 | 13822-CAGATTATATTGATTCGCACAAC | 303-ATAGAACTGATGAAGCTAAGGC | 55 | 75 |
| Numt-1-1 | GATTACGCTGTTATCCCTAAG | GGTGTAACTAGAATGATACAGGT | 51 | 65 |
| Numt-1-2 | CGGTTTGAACTCAGATCATGTAAG | GAAGGTGTAACTAGAATGATACAGGT | 57 | 78 |
| Numt-2-1 | TGCTACCTTTGCACGGTC | AGGTGAGATAAGTCGTAACATAGT | 53 | 54 |
| Numt-2-2 | TAAACTCTATAGGGTCTTCTCG | CTAGAATGATACAGGTTAGGCT | 53 | 70 |

(Numt-1-2 and Numt-2-2, Table 1) to amplify these regions and obtained the same result. These non-targeted bands were purified and sequenced, and obtained sequences belonging to mitochondria via the analysis of NCBI blast. Due to differences in mutation rates or freedom from selection pressure, they had some degree of base differences compared to corresponding mito-genome sequences.

## Sequence analysis and annotation

DNA SeqMan program, which is included in the Lasergene software package (DNASTar Inc. Madison Wis), was used to assemble sequences to obtain the complete mitochondrial genome. PCGs and rRNAs were identified by comparing *E. splendens* to *B. germanica* and *B. bisignata* [20, 21]. Most of the tRNA and their secondary structure inferences were conducted using the online server ARWEN (http://mbio-serv2.mbioekol.lu.se/ARWEN/) [24]. The tRNA-Ile, tRNA-Phe and tRNA-Leu were not identified by ARWEN; they were manually checked by referring to secondary structural models of other blattaria insects. The mitogenomic map was depicted with SeqBuilder (http://www.dnastar.com). The A+T content of the nucleotide sequence and relative synonymous codon usage (RSCU) were calculated using MEGA 5.2 [25]. The AT skewness was calculated according to the following formula: AT skew = [A-T]/[A+T], and the GC skewness was calculated according to the following formula: GC skew = [G-C]/[G+C] [26].

## Divergence dating analysis

There were two Numts found in *E. splendens*, namely Numt-1 and Numt-2. Through sequence alignment by MEGA 5.2, Numt-1 corresponded to partial lrRNA, and Numt-2 was similar to partial lrRNA and its neighboring tRNA-Val of the *E. splendens* mito-genome. We performed divergence date analyses based on the aligned sequences of Blattodea, two mantises and two outgroups (S1 Table). The molecular clock was calibrated using three minimum age constraints based on cockroach fossils, as shown in Table 2. A relaxed molecular-clock model was used for this study with the program BEAST 1.6.1 [27]. Rate variation was modeled among branches using uncorrelated lognormal relaxed clocks [27]. A Yule speciation process was used for the tree prior and posterior distributions of parameters, including the tree, were

**Table 2. Fossils used for estimation of divergence time of Numts in the analysis.**

| species | Age (Ma) | Calibration Group | Reference |
|---|---|---|---|
| *Valditermes brennae* | 130.8 | *Cryptocercus* + Isoptera | [29] |
| *"Gyna" obesa* | 57.7 | Blaberidae | [30] |
| *Cratomastotermes wolfschwenningeri* | 113 | termites | [31] |

estimated using MCMC sampling [28]. Two independent runs (each with 4 chains) of 100 million generations were sampled every 5000 generations based on the GTR model. The tree topology was then estimated using the combined sample from the last 50 million generations of each run.

# Results

## Genome content and organization

The mitochondrial genome of *E. splendens* was 15,802 bp in length with typical circular molecules. It contained 35 mitochondrial genes: 13 PCGs, 2 rRNAs (srRNA and lrRNA), 20 tRNAs and the A+T rich region (Fig 1, Table 3). Putative secondary structures of the 20 tRNAs were shown in S1 Fig. Similar to most species' mitochondria, the coding genes of *E. splendens* mito-genome are compact, with several genes overlapping. There were five overlaps totaling 21 bp with the two longest overlaps being 7 bp between *ATP8* and *ATP6*, and 7 bp between *NAD4* and *NAD4L*. There were 15 gene spacers totaling 190 bp within the entire mitochondrial genome. The longest spacer was 93 bp between tRNA-Val and srRNA, followed by 24 bp between tRNA-Ile and *NAD2* and 22 bp between *NAD1* and tRNA-Leu. The longest spacer had a similarity of 64.6% compared to its adjacent and corresponding srRNA (S2 Fig). There were 16 areas with neither gene overlap nor intergenic spacer. Additionally, there were 20 tRNAs in the mitochondrial genome of *E. splendens*, lacking two tRNAs usually located between tRNA-Ile and *NAD2*, tRNA-Gln and tRNA-Met (Fig 1).

## Nucleotide composition and codon usage

We calculated the nucleotide composition of the mtDNA in *E. splendens* using MEGA5.2, refer to S2 Table for detailed results. The content of A+T (74.6%) was higher than G+C (25.4%). It corresponded well to the AT bias generally observed in insect mito-genomes, which ranges from 69.5 to 84.9% [32, 33].

The high A+T content and nucleotide skewness of the mitochondrial genome were also reflected in the codon use of protein-coding genes. According to the relative synonymous codon usage (RSCU) value (S3 Table, Fig 2), the occurrence of synonymous codons ending in A or T was much higher than other synonymous codons, and they accounted for 89.7% (3,330) of the total codons. The third position was A or T for all codons with RSCU values greater than 1. The six most frequently used codons were also composed of A and T, namely TTT, TTA, ATT, ATA, TAT and AAT, accounting for 40.7% of the total number of codons.

## Protein-coding genes

All PCGs of *E. splendens* used ATN as the start codon, except for *COX1*. The *COX1* gene in the *E. splendens* mito-genome used TTG as the starting codon, in agreement with other known cockroaches [5]. The stop codon was most commonly TAA in the *E. splendens* mito-genome, followed by TAG (*NAD1*). Four protein coding genes: *NAD4*, *NAD6*, *ATP6*, and *NAD3*, used incomplete TA as the stop codon.

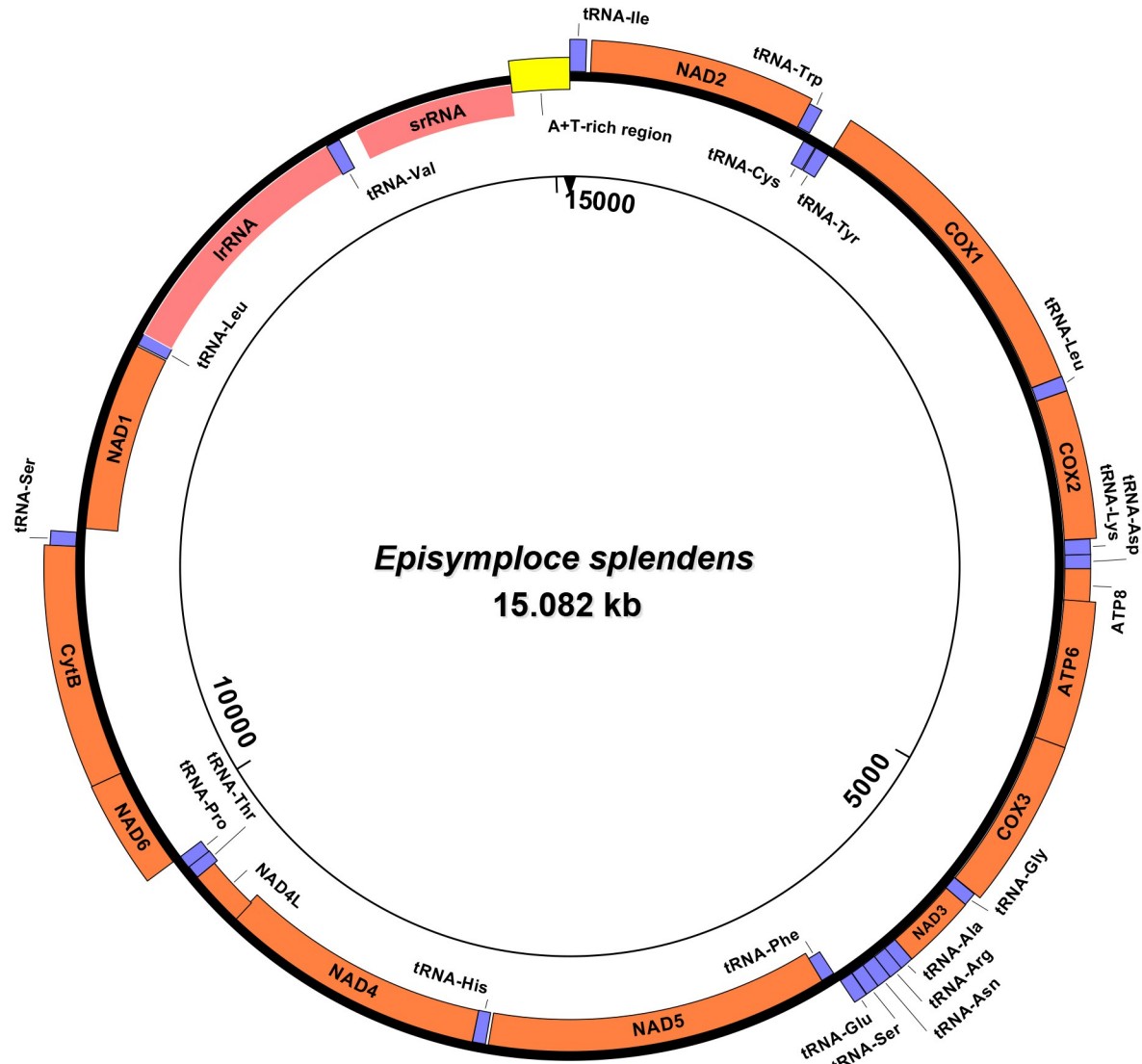

**Fig 1. Circular gene map of *Episymploce splendens* mito-genome.** Genes coded in the J-strand are inside of the circle. Gene coded in the N-strand are outside of the circle. *COX1*, *COX2* and *COX3* refer to the cytochrome C oxidase subunits; *CytB* refers to cytochrome B; ATPase6 and ATPase8 refer to ATP synthase subunits 6 and 8 genes; and *NAD1-NAD6* and *NAD4L* refer to the NADH dehydrogenase subunit 1–6 and 4Lgenes.

## A+T rich region

The non-coding region of *E. splendens* mito-genome was located between srRNA and tRNA-Ile, and it was 290 bp long with 64.8% A+T content. Two 120 bp long repeated units separated by a 7 bp interval and four Poly-A structures were found in the A+T rich region of the *E. splendens* mito-genome (Fig 3).

## Numts and its divergence time

Numts are originated from mt-genome. Numts have different mutation rates compared to their ancient mtDNA, but the pattern of their nucleotide substitution is similar to ancient mtDNA. So they can be called "fossil" markers. Numts can be used to solve some problems in

**Table 3. Annotation of *Episymploce splendens* mito-genome.**

| Gene | Grand | Location | Anticodon | Start codon | Stop codon |
|---|---|---|---|---|---|
| tRNA-Ile | J | 1..77 | TAT | | |
| NAD2 | J | 102..1148 | | ATG | TAA |
| tRNA-Trp | J | 1149..1212 | TCA | | |
| tRNA-Cys | N | 1208..1271 | GCA | | |
| tRNA-Tyr | N | 1278..1347 | GTA | | |
| COX1 | J | 1350..2885 | | TTG | TAA |
| tRNA-Leu(UUR) | J | 2887..2954 | TAA | | |
| COX2 | J | 2956..3642 | | ATG | TAA |
| tRNA-Lys | J | 3645..3715 | CTT | | |
| tRNA-Asp | J | 3717..3781 | GTC | | |
| ATPase8 | J | 3782..3940 | | ATT | TAA |
| ATPase6 | J | 3934..4613 | | ATG | TA- |
| COX3 | J | 4614..5402 | | ATG | TAA |
| tRNA-Gly | J | 5405..5468 | TCC | | |
| NAD3 | J | 5469..5821 | | ATG | TA- |
| tRNA-Ala | J | 5822..5886 | TGC | | |
| tRNA-Arg | J | 5885..5950 | TCG | | |
| tRNA-Asn | J | 5951..6020 | GTT | | |
| tRNA-Ser(AGN) | J | 6021..6085 | GCT | | |
| tRNA-Glu | J | 6088..6152 | TTC | | |
| tRNA-Phe | N | 6162..6228 | GAA | | |
| NAD5 | N | 6229..7947 | | ATT | TAA |
| tRNA-His | N | 7963..8027 | GTG | | |
| NAD4 | N | 8028..9367 | | ATG | TA- |
| NAD4L | N | 9361..9642 | | ATG | TAA |
| tRNA-Thr | N | 9643..9709 | TGT | | |
| tRNA-Pro | N | 9709..9774 | TGG | | |
| NAD6 | J | 9776..10275 | | ATT | TA- |
| CytB | J | 10276..11409 | | ATG | TAA |
| tRNA-Ser(UCN) | J | 11409..11477 | TGA | | |
| NAD1 | N | 11500..12441 | | ATA | TAG |
| tRNA-Leu(CUN) | N | 12451..12511 | TAG | | |
| lrRNA | N | 12512..13822 | | | |
| tRNA-Val | N | 13823..13893 | TAC | | |
| srRNA | N | 13987..14792 | | | |
| A+T-rich region | | 14793..15082 | | | |

'TA-' refer to incomplete stop codons.

phylogeny, such as Zischler et al used the Numt as phylogenetic outgroup to prove the origin of man [34].

There were two Numts found in *E. splendens*, namely Numt-1 (557 bp) and Numt-2 (975 bp). Comparisons with aligned mitochondrial sequence showed 87.23% homologies in Numt-1 and 76.63% in Numt-2 (S3 and S4 Figs). Some characteristics in Numts such as the deletion mutation, base substitution and insertion mutation were also found in Numt-1 and Numt-2 [35, 36].

The timescale for evolution of 25 species and Numt-1 diversification based on aligning sequences and calibrations based on three cockroach fossils is shown in Fig 4A while the time-scale for Numt-2 is shown in Fig 4B. The divergence of the lineages leading to *Blattella* and

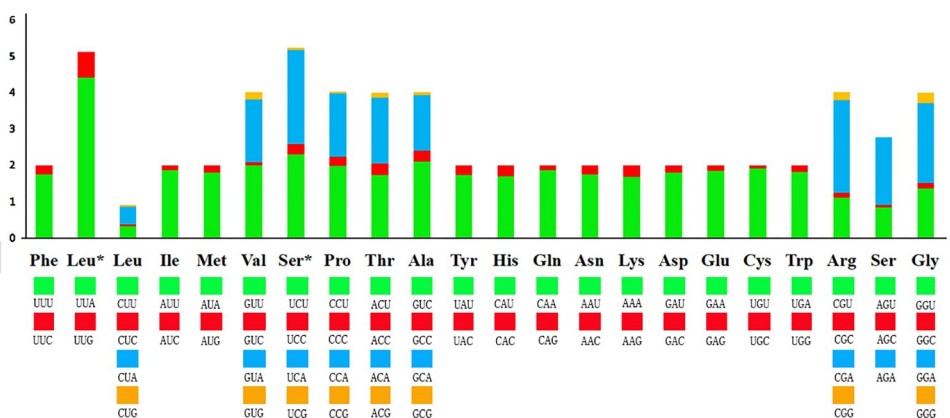

**Fig 2. Relative Synonymous Codon Usage (RSCU) in *Episymploce splendens* mito-genome.** A total of 3,711 codons for *E. splendens* mito-genome were analyzed, excluding stop codons. Leu, Leu*, Ser, and Ser* indicate trnL1 (CUN), trnL2 (UUR), trnS1 (AGN), and trnS2 (UCN), respectively.

*Episymploce* was 22.60 Ma to 36.50 Ma (95% confidence interval [CI]) in Fig 4A while the estimated age of the split between them was 15.54 Ma to 34.27 Ma (95% confidence interval [CI]) in Fig 4B. Results were similar although different aligning sequences were used to calculate the divergence time in this study (Fig 4). Numt-1 transferred from the mitochondrion to the nucleus between 14.15 Ma to 22.35 Ma (95% confidence interval [CI]), and Numt-2 were estimated to have diverged between 19.19 Ma to 24.06 Ma (95% confidence interval [CI]).

## Discussion

### Intergenic spacer

Mito-genomes typically exhibited compact arrangements, such as small gene spacing, gene overlap, or incomplete stop codons. However, the *E. splendens* mito-genome had a long

TACCTAAAAATAAAGAGTGCCCCTGTCCCCCACTA
AACTATGATTTTACCTTACCT**AAAAATAAAA**TAAG
TACAACTTCCTGCCCTTGTCCAATTTTCAACCTTTTC
AACCTTTTCAACCTTTTCAACCT**AAATAAAAAA**TA
CCGTCCCTTGTCCCCCACTGAACTATGATTTTACCT
TACCT**AAAAATAAAA**TAAGTACAACTTCCTGCCCT
TGTCCAATTTTCAACCTTTTCAACCTTTTCAACCTTT
TCAACCT**AAATAAAAAA**TACCGTCCCTTGTCCAAT
CACCA

**Fig 3. The A +T rich region sequence of *Episymploce splendens* mito-genome.** The blue and gray areas represented two 120bp-long repeated segments, respectively. The red fonts represented the poly-A structures.

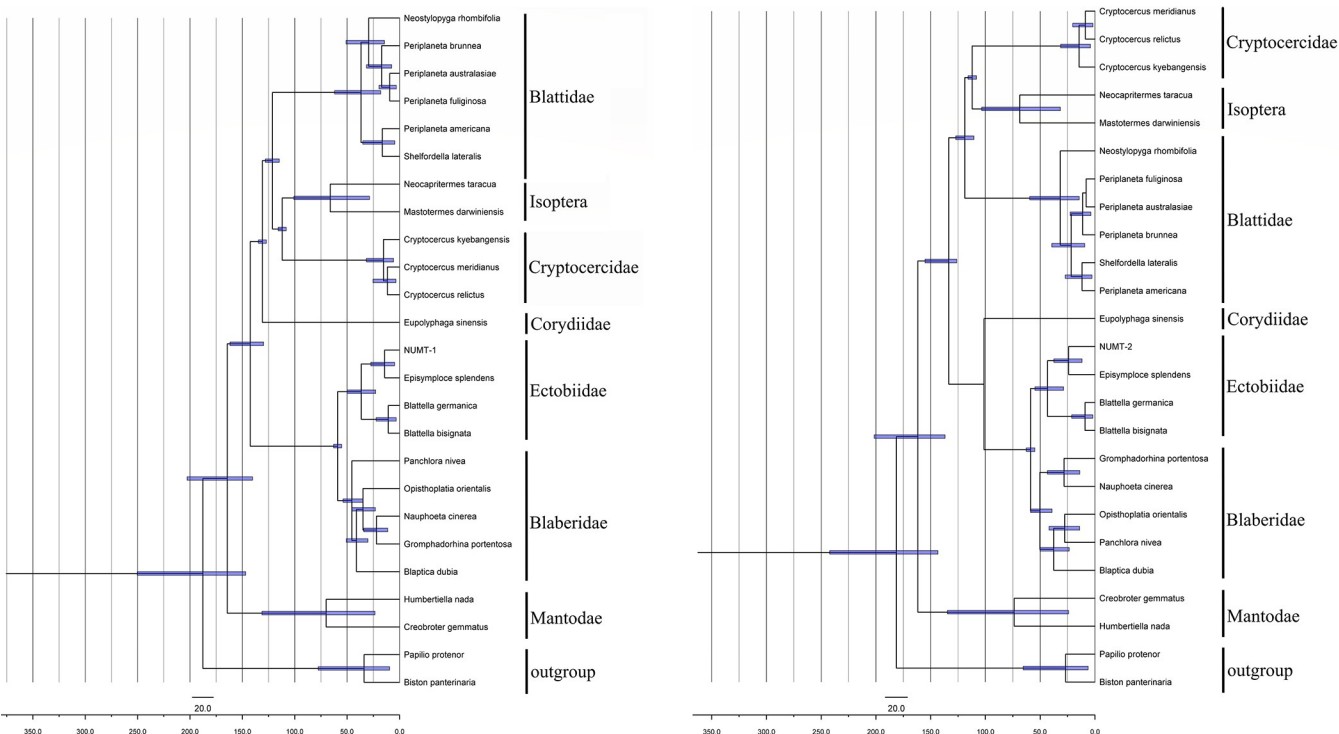

**Fig 4.** a. Phylogenetic chronogram of Numt-1 and blattodean species based on the sequence aligned with Numt-1, reconstructed using BEAST. The best-fit evolution model was determined by PartitionFinder. Scale bar estimates age in millions of years and blue bars represent 95% highest posterior density intervals for the node ages. *Papilio protenor* and *Biston panterinaria* were employed to root the tree as outgroups. b. Phylogenetic chronogram of Numt-2 and blattodean species based on the sequence aligned with Numt-2, reconstructed using BEAST. The best-fit evolution model was determined by PartitionFinder. Scale bar estimates age in millions of years and blue bars represent 95% highest posterior density intervals for the node ages. *Papilio protenor* and *Biston panterinaria* were employed to root the tree as outgroups.

intergenic spacer region (93 bp) located between tRNA-Val and srRNA, which is the longest intergenic spacer region in cockroach mito-genomes reported. The long intergenic spacers in mito-genomes have been reported in some Hymenopteran [37, 38], Hemipteran [39, 40], Dictyopteran [41] and Coleopteran insects [6, 42, 43]. There are two commonly posited evolutionary mechanisms for the origin of mitochondrial intergenic spacers, the duplication/random loss model and slipped-strand mispairing [6, 41]. We could not find a homologous sequence with both ends in this intergenic spacer, thus its formation is difficult to explain by slipped-strand mispairing [6, 44]. A similar long intergenic spacer was reported in a blattarian insect mito-genome, *Blaptica dubia* (71-bp between tRNA-Gln and tRNA-Met) [41], and the duplication/random loss model was used to explain the formation of this intergenic spacer [6, 41, 45, 46]. We suggested that this intergenic region may be derived from the replication of the 3' end of the srRNA when the DNA double helix unraveled, followed by random loss of partial duplicated gene, and then the residues formed the 93-bp remaining intergenic spacer in *E. splendens* (Fig 5).

Animal mito-genomes were generally considered to be economic and optimized for rapid replication and transcription [47]. Therefore, mitochondrial evolution had traditionally been regarded as favoring genome size reduction [3, 48, 49], possibly by eliminating intergenic spacers [50]. Eliminating nonfunctional intergenic spacers in mitochondrial evolution was important in the highly reduced and efficient mito-genomes [43]. But with the discovery of more large intergenic spacers in mito-genomes, as several containing additional origin of replication

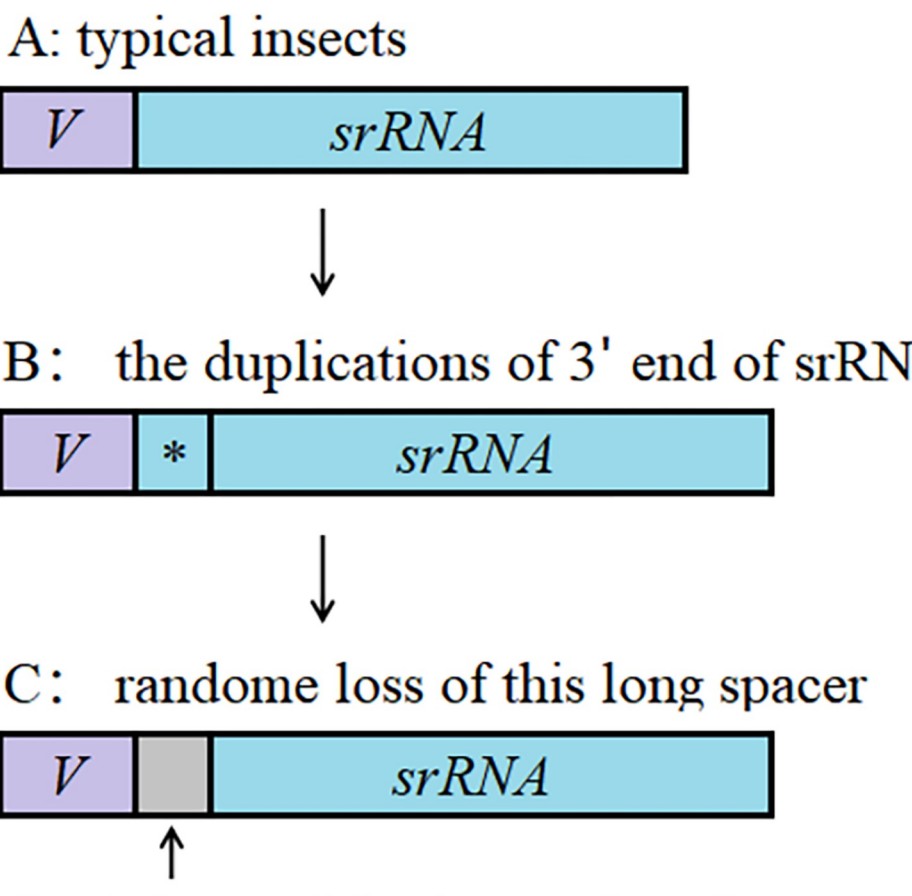

**Fig 5. Putative formation mechanism of the intergenic spacer in *Episymploce splendens* mito-genome.** The Randomly copied fragment was marked with *.

(*Apis mellifera*, *Triatoma dimidiata*, *Bombus ignitus*) [33, 39, 51], tandem repeat units (*Triatoma dimidiata*, *Pyrocoelia rufa*) [39, 42], or even open reading frames retained (*Triatoma dimidiata*) [39], whether these long spacer regions were functional was controversial [39, 52]. We analyzed the large intergenic spacers in the mito-genomes of several insects, comparing the length and similarity of these large intergenic spacer sequences with their root sequences (S4 Table) [6, 40, 41]. We found these spacer sequences were usually shorter than their root sequences and the greater the length difference between these spacer and their root sequences, the lower the similarity between them. It implied that these mito-genomes lost partial duplicated nucleotides in mitochondrial evolution. In addition, large intergenic spacers found in several related species were not homologous, implying their origin occurred independently after species differentiation [6]. This suggests that these long intergenic spacers did not confer an evolutionary advantage, and they were gradually deleted during mitochondrial evolution. Consequently, we consider that the mito-genome may continue evolving towards compact arrangement and the long intergenic region may gradually decrease or even disappear during mito-genome evolution. However, these discoveries of large intergenic spacers in insect mito-genomes contribute to species identification, and also provide valuable information for the study of the evolution of insect mitochondrial genomes.

**Table 4. The RSCU values of CAU (H) and AUU (I) in known cockroaches.**

| Cockroaches | CAU (H) | AUU (I) | GenBank Accession number |
|---|---|---|---|
| *Episymploce splendens* | 1.69 | 1.86 | OK094023 |
| *Blattella germanica* | 1.54 | 1.83 | NC_012901.1 |
| *Blattella bisignata* | 1.57 | 1.63 | NC_018549.1 |
| *Panchlora nivea* | 1.57 | 1.77 | NC_030002.1 |
| *Shelfordella lateralis* | 1.63 | 1.63 | NC_030003.1 |
| *Nauphoeta cinerea* | 1.49 | 1.64 | NC_035052.1 |
| *Cryptocercus meridianus* | 1.51 | 1.49 | NC_037496.1 |
| *Periplaneta australasiae* | 1.44 | 1.7 | NC_034841.1 |
| *Neostylopyga rhombifolia* | 1.45 | 1.66 | NC_034842.1 |
| *Periplaneta brunnea* | 1.42 | 1.71 | MG010455 |
| *Blaptica dubia* | 1.55 | 1.54 | NC_029224.1 |
| *Periplaneta americana* | 1.36 | 1.71 | NC_016956.1 |
| *Gromphadorhina portentosa* | 1.45 | 1.42 | NC_030001.1 |
| *Periplaneta fuliginosa* | 1.43 | 1.73 | NC_006076.1 |
| *Opisthoplatia orientalis* | 1.35 | 1.6 | NC_029225.1 |
| *Cryptocercus relictus* | 1.21 | 1.57 | NC_018132.1 |
| *Cryptocercus kyebangensis* | 1.33 | 1.6 | NC_030191.1 |
| *Eupolyphaga sinensis* | 0.93 | 1.64 | NC_014274.1 |

## tRNAs deletion

Mitochondrial gene content, arrangement and composition were highly conserved, and mito-genomes typically contained 37 coding genes [3, 53]. However, some exceptions, such as gene duplications, deletions or rearrangements, were found in some species [54–56]. Although tRNA deletion is unusual, increasing cases have been reported, such as three amphibians species [54, 57], three reptile species [58, 59], one crustacean species [60], one Hemiptera insect and one Coleoptera insect [61], four Psocoptera insects [10, 62] and three Mantode*a* insects [9]. In this study, two tRNAs, tRNA-Gln and tRNA-Met, were absent in *E. splendens* mito-genome. The tRNA deletions was only found in the *E. splendens* mito-genome (this study) in all Blattaria mito-genomes reported and the tRNA deletion events reported in previous research also scattered in different clade branches, therefore, we consider tRNA deletions appear to be separate events occasionally occurring in some species or evolutionary branches. The deletion of functional genes was obviously disadvantageous for species, and the mechanism of deletion is still unclear.

The organism may have a functional replacement to cope with the loss of tRNAs. Two mechanisms were proposed for this functional compensation. The first mechanism where tRNAs from the cytosol are imported into mitochondria has been confirmed, as aminoacyl-tRNA synthetases being imported from the cytosol into mitochondria [63], and functional tRNA-Lys encoded in the nuclear genome being imported into marsupial mitochondria [64]. The second compensation mechanism for the missing tRNAs is 'superwobble', where a tRNA with an unmodified U in the wobble position reads all four nucleotides in the third codon position [65]. In our study, the loss of tRNA-Gln and tRNA-Met can be compensated with their first and second codons matching His and Ile, respectively, and the anticodon swing site U for His and Ile. We calculated the Relative Synonymous Codon Usage (RSCU) values of CAU (H) and AUU (I) in reported cockroach mito-genomes, and discovered that *E. splendens* mito-genome had the highest RSCU values (Table 4), indicating that *E. splendens* might be compensated through tRNA superwobble. Regardless, we cannot exclude tRNA import from

the cytosol in *E. splendens* mitochondrion. Therefore, more studies are needed to confirm the compensation mechanism for the absence of tRNAs in the mito-genome of *E. splendens*.

## Conclusion

In this study, we sequenced and annotated the *E. splendens* mito-genome. Two tRNAs (tRNA-Gln, tRNA-Met) were lost and a long intergenic region between tRNA-Val and srRNA (93 bp, with a 64.6% similarity with its corresponding srRNA) was also found in *E. splendens* mito-genome. The duplication/random loss model may account for the origin of this long intergenic spacer. We also found two Numts, Numt-1 and Numt-2 transfered from mitochondrion to nucleus at about 14.15 Ma to 22.35 Ma, and 19.19 Ma to 24.06 Ma respectively.

## Supporting information

**S1 Table. GenBank accession numbers for divergence date analyses.**
(DOCX)

**S2 Table. Nucleotide composition in different regions of *Episymploce splendens*.**
(DOCX)

**S3 Table. Relative Synonymous Codon Usage (RSCU) for PCGs of *Episymploce splendens*.**
(DOCX)

**S4 Table. The length and similarity of these large intergenic spacers with their original genes in some insects.** The similarity was calculated by DNAMAN.
(DOCX)

**S1 Fig. Putative secondary structures of the 20 tRNA genes identified in *Episymploce splendens*.**
(TIF)

**S2 Fig. Aligement result of the long intergenic spacer and srRNA of *Episymploce splendens* by MEGA.**
(TIF)

**S3 Fig. Similarity comparison results of Numt-1 and mito-genome of *Episymploce splendens* by DNAMAN.**
(TIF)

**S4 Fig. Similarity comparison results of Numt-2 and mito-genome of *Episymploce splendens* by DNAMAN.**
(TIF)

## Acknowledgments

We sincerely appreciate Natural History Museum of Sichuan University for the sample collection. Thanks Dr. Megan for language help.

## Author Contributions

**Conceptualization:** Jinnan Ma, Xiuyue Zhang.

**Funding acquisition:** Xiuyue Zhang.

**Investigation:** Lin Yan, Zhenzhen Hou, Hongmei Wang, Jie Gao, Chenjuan Zeng, Qin Chen.

**Methodology:** Jinnan Ma.

**Project administration:** Hongmei Wang.

**Resources:** Lin Yan, Jie Gao, Chenjuan Zeng, Bisong Yue.

**Software:** Jinnan Ma.

**Writing – original draft:** Lin Yan, Zhenzhen Hou, Hongmei Wang, Jie Gao, Qin Chen.

**Writing – review & editing:** Lin Yan, Bisong Yue, Xiuyue Zhang.

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
