## [Decision Letter · Decision Letter 0]

5 Jul 2021

PONE-D-21-15397

Complete mitochondrial genome of Episymploce splendens (Blattodea: Ectobiidae): a large intergenic spacer and lacking of two tRNA genes

PLOS ONE

Dear Dr. Zhang,

Thank you for submitting your manuscript to PLOS ONE. After careful consideration, we feel that it has merit but does not fully meet PLOS ONE’s publication criteria as it currently stands. Therefore, we invite you to submit a revised version of the manuscript that addresses the points raised during the review process.

We look forward to receiving your revised manuscript.

Kind regards,

Tzen-Yuh Chiang

Academic Editor

PLOS ONE

Journal Requirements:

Reviewers' comments:

Reviewer's Responses to Questions

**Comments to the Author**

1. Is the manuscript technically sound, and do the data support the conclusions?

Reviewer #1: Partly

Reviewer #2: Yes

2. Has the statistical analysis been performed appropriately and rigorously? 

Reviewer #1: No

Reviewer #2: Yes

3. Have the authors made all data underlying the findings in their manuscript fully available?

Reviewer #1: No

Reviewer #2: Yes

4. Is the manuscript presented in an intelligible fashion and written in standard English?

Reviewer #1: Yes

Reviewer #2: Yes

5. Review Comments to the Author

Reviewer #1: In this manuscript, the authors sequenced the mitochondrial genome of the cockroach species Episymploce splendens. The mitogenome lacks two tRNAs and has a long intergenic spacer (93 bp long) between tRNA-Val and srRNA. Then, the authors conducted phylogenetic analyses for Blattodea relying on 13 protein coding genes.

I have several concerns regarding this manuscript, from structure issues (especially discussion inserted within the results) to a lack of methodological details and an inadequate (according to me) phylogenetic section.

The authors report the first mitogenome of Blattodea and relatives that lack a couple of tRNAs. This is interesting data to investigate the molecular evolution of mitogenomes in insect. I regret, however, that the authors do not provide the sequences yet, which is, for a reviewer, highly frustrating. I would have love to see the data, look at the annotation and so on. I consider that, for such a manuscript, it should be mandatory to provide this kind of information for the review process. Also, I was curious about the software (ARWEN) used to annotate the mitogenome and, more precisely, how others software might have been used to confirm the results of the authors. This is especially troubling because ARWEN did not recognize 3 tRNAs that the authors had to identify manually. How likely is it that the 2 tRNAs reported here as lacking could have also been missed by ARWEN and the authors?

Note that I am not relevant enough to discuss further the quality and completion of the data and analyses provided regarding the composition of the mitogenomes or the evolutionary mechanisms. Nevertheless, I am a bit skeptical about the dating analyses with NUMTs. I wonder if those analyses really make sense (i.e. aligning mitochondrial sequences with a NUMt, and then trying to estimate divergence ages).

As for the phylogenetic part, I simply think that it should be removed from the manuscript and I will try to explain my position here. First, and above all, the data brought here (i.e. 1 mitogenome of a species that undoubtedly belongs to the Ectobiidae family) does not bring enough additional data or new results to justify those phylogenetic analyses. In addition, the authors did not use all the mitogenomes available and we have no idea why some taxa were used while others were disregarded. Second, the vocabulary used (i.e. mainly the word ‘basal’) leads to some very surprising sentences for phylogeneticists, which might reflect a lack of understanding of phylogenetic analyses or at least of the phylogenetics of Blattodea. Third, there are several parameters that are not provided in the manuscript or methodological choices that should be explained in more details (e.g. two Lepidoptera as outgroups? ‘unmatched’ regions removed?). Finally, the literature cited is not up to date and biased with, for instance, a study relying on a single marker cited while the most recent studies are disregarded. Overall, this part of the manuscript is flawed and does more harm than good to this manuscript.

Reviewer #2: This paper adds significant new data and knowledge to Blattodea mitochondrial genome. This paper deserves publication, but there are several issues that need to be addressed prior to publication, mainly in the phylogenetic study and fossil selection. Please see the attachment for details.

6. PLOS authors have the option to publish the peer review history of their article (what does this mean?). If published, this will include your full peer review and any attached files.

Reviewer #1: No

Reviewer #2: No

---

## [Author Response · Author response to Decision Letter 0]

20 Sep 2021

Reviewer 1:

In this manuscript, the authors sequenced the mitochondrial genome of the cockroach species Episymploce splendens. The mitogenome lacks two tRNAs and has a long intergenic spacer (93 bp long) between tRNA-Val and srRNA. Then, the authors conducted phylogenetic analyses for Blattodea relying on 13 protein coding genes.

I have several concerns regarding this manuscript, from structure issues (especially discussion inserted within the results) to a lack of methodological details and an inadequate (according to me) phylogenetic section.

The authors report the first mitogenome of Blattodea and relatives that lack a couple of tRNAs. This is interesting data to investigate the molecular evolution of mitogenomes in insect. I regret, however, that the authors do not provide the sequences yet, which is, for a reviewer, highly frustrating. I would have love to see the data, look at the annotation and so on. I consider that, for such a manuscript, it should be mandatory to provide this kind of information for the review process. Also, I was curious about the software (ARWEN) used to annotate the mitogenome and, more precisely, how others software might have been used to confirm the results of the authors. This is especially troubling because ARWEN did not recognize 3 tRNAs that the authors had to identify manually. How likely is it that the 2 tRNAs reported here as lacking could have also been missed by ARWEN and the authors?

Note that I am not relevant enough to discuss further the quality and completion of the data and analyses provided regarding the composition of the mitogenomes or the evolutionary mechanisms. Nevertheless, I am a bit skeptical about the dating analyses with NUMTs. I wonder if those analyses really make sense (i.e. aligning mitochondrial sequences with a NUMt, and then trying to estimate divergence ages). As for the phylogenetic part, I simply think that it should be removed from the manuscript and I will try to explain my position here. First, and above all, the data brought here (i.e. 1 mitogenome of a species that undoubtedly belongs to the Ectobiidae family) does not bring enough additional data or new results to justify those phylogenetic analyses. In addition, the authors did not use all the mitogenomes available and we have no idea why some taxa were used while others were disregarded. Second, the vocabulary used (i.e. mainly the word ‘basal’) leads to some very surprising sentences for phylogeneticists, which might reflect a lack of understanding of phylogenetic analyses or at least of the phylogenetics of Blattodea. Third, there are several parameters that are not provided in the manuscript or methodological choices that should be explained in more details (e.g. two Lepidoptera as outgroups? ‘unmatched’ regions removed?). Finally, the literature cited is not up to date and biased with, for instance, a study relying on a single marker cited while the most recent studies are disregarded. Overall, this part of the manuscript is flawed and does more harm than good to this manuscript.

Response:

We really appreciate your valuable time on reviewing our manuscript. We provide the complete mt-genome to Genbank and obtained the GenBank accession number (OK094023) although we set up the release date. Whatever, we provide the Genbank annotation format file to facilitate the review, and the annotation information of this mitochondrial genome is also presented in Table 3. Actually, except for ARWEN, we used tRNA-SE to identify tRNAs and obtained identical result. The tRNAs we did not identify by softwares were identified by manually, through reference to secondary structure models for those genes from other blattarian insects (Ma et al., 2017; Xiao et al., 2012). We drew 20 tRNAs in hand, and could not find the remaning 2 tRNAs, tRNA-Gln and tRNA-Met. tRNA-Gln and tRNA-Met located between tRNA-Ile and NAD2, and they were typically 150bp long. However, this site was only 24bp long in E. splendens and we also could not found their secondary structures here. We designed differtrnt primers to amplify this area and obtain same amplification sequences, so we think our experiment result was correct. Although tRNA lacking was rare, other species had been found in previous studies. For example, Limnonectes bannaensis lost 4 tRNAs, respectively tRNA-Ala, tRNA-Asn, tRNA-Cys and tRNA-Glu. Gegeneophis ramaswamii lost tRNA-Phe in mt-genome. Urochela quadrinotata Reuter lost 2 tRNAs, respectively tRNA-Ile and tRNA-Gln. Platysternon megacephalum lost tRNA-Thr. So we supported the 2 tRNAs were missing in Episymploce splenden mt-genome.

Nuclear mitochondrial pseudogenes (Numts) are DNA fragments transferred from mitochondria to the nucleus, which may become non-functional due to mutation and loss of selective pressure. Since the mutation rate in the nucleus is lower than that of the mitochondria, the divergence time can be calculated according to the sequence differences between them and ancient mtDNA. These Numts are equivalent to fossils of mitochondrial DNA left in nuclear DNA, and the divergence time of species can be estimated based on identical or similar Numts in related species. So they can be called “fossil” markers. Numts can be used to solve some problems in phylogeny, such as Zischler et al (1995) used the Numt of D-loop as phylogenetic outgroup to prove the origin of man, Lou et al(2013) used 2 Numts as “fossil” markers to trace back the speciation time of two sibling fig wasp species. So we consider the research on Numts is meaningful.

In phylogenetic analyses, the absence of some species may lead to uncertain inferences for phylogenetic relationships (Tomita et al., 2011). Increased samplings can provide more complete and reliable information for phylogenetic analysis. In this study, we amplified the entire mt-genome sequence of Episymploce cockroaches for the first time. Performing phylogenetic trees with mt-genomes was used to confirm the taxonomic status of E.splendens. At the same time, our phylogenetic analysis also further some phylogenetic relationships, and the reviewer 2 suggested us to use more phylogenetic analysis methods. Therefore, we had retained this part. However, as your comment, this part was not the highlight in our manuscript, so we tried to simplify it and….. Blattaria contains 4,600 described species, but only limited complete mitochondrial genomes have been reported. At present, there are still little known complete mt-genomes of most cockroaches. Many cockroaches are unmentioned since their mt-genomes are still unknown. Besides, we are sorry for using the word “basal”. We had changed “basal” in the whole manuscript (page 2, line 40; page 4, line 70; page 16, line 286; page 26, line 493).

Phylogenetic trees that we build are typically unrooted trees and we need some outgroups to root the tree to reflect the ancestral relationships. Outgroup species are usually closely related to and can be separated from the species to be studied. Both Lepidoptera and Orthopteroidea belong to Pterygota, and have closely relationship, so we chose the two Lepidoptera as outgroups. Besides, previous study on the phylogenetic relationships among cockroaches also chose same two Lepidoptera as outgroups (Gong et al., 2018). 

“Unmatched” regions couldn't provide any useful information in phylogenetic analysis, so they were usually deleted during sequence alignment. About the cited literatures, thanks again for your advice and sorry for these careless mistakes. Now we have carefully read and considered your comments, and we had cited the latest litaratures in the manuscript. The following are point by point responses to your comments:

1.Page 2, line 32. What is Orthopterodea? Is it a commonly accepted term for the clade you have in mind?

Response:

Thanks for your suggestion and sorry for the careless mistake. We have changed “Orthopterodea” to “Orthopteroidea”.

2. Page 2, line 33. It reads as if you were the first to propose this mechanism. Please rephrase.

Response:

Thanks for your suggestion. We changed “proposed” to “supported”.

3.Page 2, line 38. I do not understand what you mean by 'molecular fossils'.

Response:

Nuclear mitochondrial pseudogenes (Numts) are DNA fragments transferred from mitochondria to the nucleus, which may become non-functional due to mutation and loss of selective pressure. Since the mutation rate in the nucleus is lower than that of the mitochondria, the divergence time can be calculated according to the sequence differences between them and ancient mtDNA. These Numts are equivalent to fossils of mitochondrial DNA left in nuclear DNA, and the time of divergence can be estimated based on identical or similar Numts in related species (Lopez et al., 1994; Thalmann et al., 2004). Besides, Numts can be used to solve problems in phylogeny, such as Zischler et al(1995) used the Numt of D-loop as phylogenetic outgroup to prove the origin of man, Lou et al(2013) used 2 Numts as “fossil” markers to trace back the speciation time of two sibling fig wasp species. 

4.There is no 'basal' position. This kind of terms should be banned. Here Blaberoidea is sister-group to Blattoidea, and none of those clades is basal'.

Response:

We changed “basal” in the whole manuscript (page 2, line 40; page 4, line 70; page 16, line 286; page 26,line 493). 

5.Page 2, line 40. Well, this result is not new and has been discussed at length in the literature. I do not think it should be a highlight of your study.

Response:

Thank you for your valuable advice. We have removed "Isoptera should be classified as a family within Blattodea and Mantodea as the sister clade to (Isoptera + Blattodea)." .

6.Page 2, line 44. Well, every new data in this regard is welcome for sure but to say that your study bring "important data for further studies on (.) the phylogenetic relationships of Blattodea" is a huge overstatement. Several tens of mitogenomes are availlable and you provide the community with an additional one.

Response:

Thanks for your suggestion. We changed “These discoveries could deepen our understanding of mitochondrial genome of insects, and increase the clarity of Blattodea phylogenetic relationships and contribute to the study of insect mito-genome evolution” to “Our study provided useful data for further studies on insect mito-genome evolution and the phylogenetic relationships of Blattodea.”.

7.Page 3, line 64. Given the species richness of Blattodea, I would say 'a few' rather than 'many (i.e. ca. 20 species, out of >4000)

Response:

Thank you for your valuable advice. We have changed “many” to “a few”.

8.Page 4, line 73. Remove?

Response:

 We haved removed “Episymploce”.

9.Page 4, line 77.Sequenced

Response:

We have changed “determined” to “sequenced” in our revised manuscript.

10.Page 4, line 82. spacer and not spacers if you found only one as the abstact suggests.

Response:

Thanks for your advice and sorry for the careless mistake. We changed “spacers”to “spacer”.

11.Page 5, line 90. I believe five species of Episymploce are known from China mainland (and I guess more await description). Can you give the readers some details on the way you identified the specimens?

Response:

Thanks for your suggestion. We added some details that we identified the specimens. The Episymploce species have typical characteristics of adult cockroaches such as two sets of wings, sclerotized tegmina and wider membranous hindwings. Besides, the pronotum of E. splendens is black and one-orange patch is on each side of the lateral border. The front and hind wings are fully developed and through the end of the abdomen. Supra-anal plate of male genitalia is asymmetrical (page 5, line 90).

12.Page 5, line 98. Remove

Response:

Thanks for your advice. We delete “it”.

13.Page 5, line 99.Which muscle tissue, please clarity.

Response:

Thanks for your advice. We have added the location that muscle tissue was sampled in the manuscript.

14.Page 5, line 109. I do not understand this sentence. How specific primers have been designed, besides comparing the conserved regions of the two Blattella species mentioned before?

Response:

Thank you for your careful work. We firstly designed primers using the conserved regions of two Blattella species mito-genomes to amplify some regions of E. splendens mito-genome. And then the sequenced regions were used to design specific primers to amplify the remaining sequences of E. splendens. Primers were designed by the software Primer Premier 5.

15.Page 6, line 119. what do you mean here by 'estimated"? Please clarify.

Response:

Through 1.0% agarose gel electrophoresis, we can preliminarily estimate if the PCR pruduct was the target band we want, such as if it was the right size.

16.Page 8, line 126. We do not know to which sequences this 'these' refers to? Is a sentence missing?

Response:

Thank you for your careful work. There is no sentence missing. To avoid ambiguity, we changed “these sequences” to “The sequences” in our revised manuscript.

17.what about Mitos or other software classically used for this kind of purpose?And if ARWEN did not find three tRNAs, how sure can we be about the two others that you consider as missing for this species?

Response:

Actually, except for ARWEN, we used tRNA-SE to identify tRNAs and obtained similar result. The tRNAs we did not identify by softwares were identified by eye, through reference to secondary structure models for those genes from other blattarian 

insects (Ma et al., 2017; Xiao et al., 2012). We drew 20 tRNAs in hand, and could not find the remaning 2 tRNAs, tRNA-Gln and tRNA-Met. tRNA-Gln and tRNA-Met lacated between tRNA-Ile and NAD2, and they were typically 150bp long. However, this site was only 24bp long in E. splendens. We designed differtrnt primers to amplify this area and obtain same amplification sequences, so we think our experiment result was correct. We could not found their secondary structures here. Although tRNA lacking was rare, other species had been found in previous studies. For example, Limnonectes bannaensis lost 4 tRNAs, respectively tRNA-Ala, tRNA-Asn, tRNA-Cys and tRNA-Glu. Gegeneophis ramaswamii lost tRNA-Phe in mt-genome. Urochela quadrinotata Reuter lost 2 tRNAs, respectively tRNA-Ile and tRNA-Gln. Platysternon megacephalum lost tRNA-Thr. So we supported the 2 tRNAs were missing in Episymploce splenden mt-genome.

18.Page 9, line 158. I guess this is not correct as your sampling includes Caelifera and Ensifera.

Response:

Thanks for your advice. We changed “8 grasshopper” to “4 grasshoppers, 4 crickets”.

19.Choosing two Lepidoptera as outgroups should be better explained. I think it is a bad choice and the relationships found regarding Orthoptera notably make me think that I might be right.

Response:

Phylogenetic trees that we build are typically unrooted trees and we need some outgroups to root the tree to reflect the ancestral relationships. Outgroup species are usually closely related to and can be separated from the species to be studied. Both Lepidoptera and Orthopteroidea belong to Pterygota, and have closely relationship. Species differentiation of them is 143 Ma to 250 Ma, so we chose the two Lepidoptera as outgroups. Besides, previous study on the phylogenetic relationships among cockroaches also chose same two Lepidoptera as outgroups (Gong et al., 2018). 

20.Page 9, line 166. What were these regions? How long were they? This should be stated in the manuscript. It is all the more important that we do not expect 'unaligned or unmatched' regions for the PCG！

Response:

“Unmatched” regions couldn't provide any useful information in phylogenetic analysis, so they were usually deleted during sequence alignment. After alignment in MEGA 5.2, we removed the unaligned sequences at both ends of the sequence and in the middle. Because different species had different mitochondrial sequence lengths, so the sequences we deleted were various. The remaining length of nucleotide data of 13 PCGs was 9,923bp, amino acid sequences of 13 PCGs was 10,032bp, and lrRNA+srRNA was 1,375bp.

21.Page 9, line 168. What about the models for the amino acid sequences? You must provide them as well here.

Response:

We modified our manuscript and made it more clear. In the advised manuscript, we provided the models for the amino acid sequences. The GTR+I+G model was chosen as the best-fitting model for BI analysis, the GTR model was chosen as the best-fitting model for the ML analysis.

22.What about convergence diagnostics? Did you check for convergence and if yes, how? Please give the readers all these details.

Response:

Convergence diagnostics were done after constructing the trees, but we did not show it in the manuscript at first. The log files could be generated after the construction of BI trees and we used them to conduct convergence diagnostics. In these files, we found all average standard deviations were down to 0.01. This showed the results were credible.

23.Page 10, line 181. 'in’ rather than "by. Note that the sentence is awkward because you worte that you modified the trees. I guess you mean that you use FigTree to read the tree, add support values and error bars or these kind of things, but that you did not modify the tree (i.e did not modify the phylogenetic relationships).

Response:

Thanks for your careful work. We changed “by” to “in”. Yes, we used Figtree to read our trees, add support values and GenBank accession number to the node and to the corresponding species, respectively. We did not modify the tree, so we changed “modified” to “visualized”.

24.Page 14, line 252. I don't get it. What are melois? How did you compute the values for this Figure 2?

Response:

Thanks for your advice and sorry for the careless mistake. It should be “E. splendens” instead of “melois”. We put the sequences of 13 PCGs (expect for the stop codon) to MEGA5.2 to compute the RSCU. We made corresponding revision in our revised manuscript.

25.The results section if a mx of results and discussion (as the numerous reference cited in the result section show)

Response:

Thanks for your careful work. We modified this part and remove “The TA stop codon is common in insect mitochondrial genomes and it has been suggested that these incomplete stop codons can be completed by adding A during transcription without after translation [2] [46]”.

26.all Lepidoptera I think and I do not see what is the point of discussing this as it was not found in Episymploce splendens.

Response:

Thanks for your advice. We had removed “The initiation codons of COI genes were various in different insect species, with typical COI triple start codons including ATG and ATC. However, non-ATN trinucleotide (CGA, TTG) codons, tetranucleotides (ATAA, TTAA, and ATTA) and hexanucleotides (ATTTAA) have been found as the start codons of COI genes in some species [40] [41] [42] [43] [44] [45]”. 

27.Again this belongs to the discussion section, not the results

Response:

Thanks for your careful work. We remove “The A+T rich region plays an important role in mitochondrial transcription and replication [47] [48]. Since the control region does not code for genes, the degree of variation in the control region is high and the length varies greatly due to its non-coding characteristic [49]. For example, the non-coding region in insects varies from 70 bp in Ruspolia dubia to 9-13 kb in Pissodes spp. [50] [51]” “Repeated units in the control regions were also found in different blattarian insects. However, these repeated sequences had low similarity in various cockroaches and thus may have different origins [4]”.

28.The word basal should be removed from this manuscript. It can be done quite easily. For instance, Mantodea is not basal to... it is sister-group to Blattodea.

Response:

Thank you for your valuable advice. We revised the whole manuscript (page 2, line 40; page 4, line 70; page 16, line 286; page 26,line 493).

29.Page 15, line 280. I'd say polyphyletic.

Response:

We changed “paraphyletic” to “polyphyletic”.

30.Page 16, line 297. I do not think that it is an appropriate reference. Other works specifically dealt with support values and compare PP and bootstrap values. Those should be cited instead.

Response:

Thanks for your valuable advice. We refer to relevant literatures and found this is not an absolute conclusion (Cummings et al., 2008), whether bootstrap or posterior probability values are higher depends on the data in support of alternative topologies. So we remove the reference.

31.Page 17, line313. I do not understand this sentence. 

Response:

We changed “We found that the deletion mutation, base substitution and insertion mutation conformed to Numt-1 and Numt-2” to “some characteristics such as the deletion mutation, base substitution and insertion mutation were also found in Numt-1 and Numt-2”.

32.What is this optimal partitioning scheme. What are the parameters of this analysis? What is the rationals for the taxonomic sampling? There is no CI for crown-Dictyoptera and crown-Blattodea. Is it because those ages were fixed (or very very constrained) in the analyses. If yes, then the slight differences in age estimates between the two topologies might reflect these very strong constraints rather than a signal from the data.

Response:

Thanks for your advice. This is our poor description and we just want to descript that the best-fit evolution model was determined by PartitionFinder. We changed “An optimal partitioning scheme was determined by PartitionFinder” to “The best-fit evolution model was determined by PartitionFinder”. Beside, the parameters we used were substitution model (GTR), base frequencies (empirical), site heterogeneity model (Gamma), molecular clock model (estimate relaxed clock: uncorrelated lognormal), tree prior (yule process), fossils and MCMC (length of chain:50,000,000; echo state to sereen every: 1000). The rationals of taxonomic sampling based on known fossils and the kinship between species. According to another reviewr’s suggestion, we reselected fossils to calculate the divergence time and produce confidence intervals. We found in Fig 5(b), the clade (Eupolyphaga sinensis + Blaberoidea) did not produce CI, we guess it may be caused by lack of significance level. You can see these information in Page 10, line 191.

33.Page19, line 354. I do no really understand the rationale here. And why mentioning the realtively long intergenic spacer in Blaptica dubia? What does it suggest?

Response:

According to slipped-strand mispairing theory, mispairing involves dissociation of replicating DNA strands and then misaligned reassociation, following replication or repair lead to insertions of several repeat units. Formed tandem repeat experienced random loss and/or point mutation, only the repeat units in both ends are completely retained and the residues form the long spacer (Du et al, 2017). We could not find a homologous sequence with both ends in this intergenic spacer in E.splendens, so we think this region is not formed by this model. Besides, similar long intergenic spacer in Blaptica dubia had been reported and its formation was speculated by the duplication/random loss model. So we assumed that this long intergenic spacer in E.splendens was generated by the duplication/random loss model.

34.I think Fig. 6 is useless.

Response:

We believe that Fig. 6 is a useful illustration of the formation of the long intergenic spacer, and we keep this figure in our manuscript for readers to understand.

35.Page 20, line 384. I am not sure to understand what you mean by 'root sequences' and how they were computed. And from here starts a long section with a single reference about beetle mitogenomes. Do you have any references to support your claims or is it purely speculative?

Response:

The long spacers listed in S5 Table were originated from protein-coding genes in mt-genomes, they have a certain similarity. So we called the original sequences of protein-coding genes root sequences. MEGA5.2 was used to compare the long spacer region with the protein-coding genes to determine the root sequence. The similarity between the root sequence and long spacer was computed by the software DNAMAN. Through the length comparison, similarity comparison and spacer locations, we carefully set out our speculation in the manuscript.

36.Page 21, line 405. What were the tRNAs lost in these different species? Is it a single tRNA in each species or are there several tRNAs in some of these species?

Response:

The amount of tRNAs lost varies among species. For example, Limnonectes bannaensis lost 4 tRNAs (tRNA-Ala, tRNA-Asn, tRNA-Cys and tRNA-Glu), Gegeneophis ramaswamii lost tRNA-Phe in mt-genome, Urochela quadrinotata Reuter lost 2 tRNAs (tRNA-Ile and tRNA-Gln), Platysternon megacephalum lost tRNA-Thr.

37.Please remove this. This has been repeatedly published, discussed, with data specifically designed to address this issue, which is not the case for your study. I think the whole the phylogenetic section should be deleted.

Response:

We deleted “The Blattodea, Isoptera and Mantodea all belonged to Dictyoptera based on morphological and anatomical characteristics [11] [12]. According to molecular and morphology data, Mantodea was considered as the basal split within Dictyoptera [81] [82]. However, DeSalle et al [83] supported that Mantodea and Isoptera were sister groups within Dictyoptera. Nevertheless, more recent studies discovered that Mantodea was the sister clade to (Isoptera + Blattodea), and even indicated that Isoptera should be classified as a family (Termitidae) within Blattodea [34] [39] [84]. In our study, all results supported Mantodea was a sister clade to Blattodea (including Blattodea and Isoptera) with high support. In addition, we found Isoptera clustered into a monophyletic clade with high support (Fig 4 and S3 Fig), and this was also supported by several previous studies [9] [12] [85]. Consequently, we propose that Isoptera should be classified as a family within Blattodea”.

38.Page 23, line 446. citing here a paper listing the cockroaches from Colombia is very very weird!

Response:

We cited Djernæs (2018) instead of Velez et al (2006).

39.Page 24, line 459. is, not was

Response:

We changed “was” to “is”.

40.Page 24, line 447. this is a very biased interpretation of the situation and of the literature. Citing here a few references 20-25 years old relying on a single mitochondrial marker with a few taxa is not a good represetation of the state-of-the-art of Blattodea phylogenetics. This cannot be published as is.

Response:

Thanks for your advice. We rewrote this part and cited the recent references. We changed “Although about 4,000 species of cockroaches have been recorded, the phylogenetic relationships within the Blattodea have been debated. Blaberoidea was comprised of Blaberidae and Ectobiidae. The relationships within Blattodea, between Blaberoidea and the remaining cockroaches, have been disputed. Most researchers have supported Blaberoidea being clustered with the remaining cockroaches, while some studies supported Blaberoidea as the sister group to Blattidae. In our study, all trees strongly supported Blaberoidea being placed in the basal position of Blattodea. In addition, within Blaberoidea, we found Ectobiidae was paraphyletic while Blaberoidea was monophyletic (Fig 4 and S3 Fig), consistent with several previous studies. However, more Ectobiidae samples needed to be added in the future to confirm our finding.” to “Although about 4,000 species of cockroaches had been recorded, the relationships within Blattodea had been debated. Blattodea can be divided into three superfamilies: Blaberoidea, Blattoidea and Corydioidea. Many studies supported Blaberoidea being clustered with the Blattoidea+Corydioidea using both morphological and molecular characters , transcriptomes, gene locis, but some studies supported Blaberoidea+Corydiidae as the sister group to the remaining cockroaches, such as Lo et al (used combined dataset of 18S rRNA, histone 3 and COII), Lo et al (based on 18S rDNA and COII). In our study, we got different results. The clade ((Ectobiidae + Blaberidae)+Corydiidae) was clustered with remaining cockroaches based on lrRNA+srRNA in the BI analysis while other trees strongly supported Blaberoidea as the sister group to the clade Blaberidae+Corydiidae+Blattidae+Crypyocercidae+Isoptera. Blaberoidea is comprised of Blaberidae and Ectobiidae. In this study, within Blaberoidea, we found Ectobiidae was paraphyletic while Blaberoidea was monophyletic (Figs 4 and S3), consistent with several previous studies”.

41.Page 24, line 462. Well other studies with a more extensive taxonomic sampling already confirm the paraphyly of Ectobiidae (see for instance Evangelista et al. 2021 for I guess the latest study with several tens of nuclear loci)!

Response:

Thanks for your advice. We revised our manuscript and changed “However, more Ectobiidae samples needed to be added in the future to confirm our finding” to “In this study, within Blaberoidea, we found Ectobiidae was paraphyletic while Blaberoidea was monophyletic (Figs 4 and S3), consistent with several previous studies”.

42.Page 24, line 463. I do not understand how this can be the focus of a discussion section while you bring new data for an uncontroversial Ectobiidae species? In addition, the most recent studies do not show such huge disagreements as you suggest. 

Response:

The position of Corydiidae was controversial and it had been debated for many years. Beside, we obtained different topologies in different trees. We supported this was a valuable discussion point.

43.I wonder whether it really makes sense at all. Aligning mitochondrial sequences with Numts (that are therefore not homologous) and then try to estimate divergence ages does not make sense according to me.

Response:

Numts have a low rate of evolution, and they can be used to solve problems in phylogeny, such as Zischler et al(1995) used the Numt of D-loop as phylogenetic outgroup to prove the origin of man, Lou et al(2013) used 2 Numts as “fossil” markers to trace back the speciation time of two sibling fig wasp species. So we consider their research is meaningful.

Reviewer 2:

This paper adds significant new data and knowledge to Blattodea mitochondrial genome. This paper deserves publication, but there are several issues that need to be addressed prior to publication, mainly in the phylogenetic study and fossil selection. Please see the attachment for details.

Response:

Thank you for your careful reading of our manuscript. We have carefully read and considered your comments, and we believe that the revised manuscript has been greatly improved in comparison to the previous submission thanks to the critical comments received. The following are point by point responses to your comments:

1.Page 2, line 32. This is not first time report of mt genome rearrangement of Orthopterodea, such as Cryptocercus in Blattodea.

Response:

Thanks for your suggestion. What we emphasized here was tRNA deletions in mt genome instead of mt genome rearrangement in Orthopterodea.

2.Page 2, line 38. Delete and rewrite. The authors should highlight the two NUMTS not the phylogenetic relationships within Blattodea.

Response:

Thanks for your suggestion. We revised this part and reduced the descriptions of phylogenetic relationships within Blattodea. We changed “The phylogenetic relationships were analyzed based on the amino acid and nucleotide sequences of 13 PCGs using maximum likelihood and Bayesian inference and a similar topology was yielded, except for the position of Caelifera and Eupolyphaga sinensis (Corydiidae). Tree results supported that (Blattidae + (Corydiidae + (Cryptocercidae + Isoptera))) and Blaberoidea being placed in the basal position of Blattodea, Isoptera should be classified as a family within Blattodea and Mantodea as the sister clade to (Isoptera + Blattodea)” to “The phylogenetic results with different datasets and inference methods showed different topologies. We found ((Ectobiidae + Blaberidae)+Corydiidae) was the sister group of the remaining cockroaches in Blattodea based on lrRNA+srRNA in the BI analysis while other trees strongly supported Blaberoidea as the sister group to the clade Blaberidae + Corydiidae + Blattidae + Crypyocercidae + Isoptera. Besides, the position of Corydiidae was uncertain”.

3.Page 3, line 58. When it comes to exceptions, why not mention gene rearrangement in many insect orders, mt genome fragmentation in Anoplura and so on.

Response:

Thank you for your careful work. We had made additions to the manuscript and supplemented gene rearrangements.

4.Page 3, line 64. Some.

Response:

Thanks for your suggestion and sorry for the careless mistake. Considering the species richness of Blattodea, we had changed “many” to “a few”.

5.Page 3, line 65. The reference here is incomplete.

Response:

Thanks for your suggestion and sorry for the careless mistake. We consulted relevant literatures and supplemented it in the manuscript.

6.Delete.

Response:

Thank you for your valuable advice. We deleted "interfamily".

7.Page 4, line 68. The authors should list the latest literatures here. Especilally Bourguignon et al., 2018, they used more than 100 mt genomes to reconstruct the phylogeny of Blattodea.

Response:

Thanks for your constructive suggestion. We listed the latest literatures here, such as Evangelista et al(2018), Djernæs et al(2015). 

8.Page 4, line 77. Citation errors of references.

Response:

Thanks for your suggestion and sorry for the careless mistake. We listed Li et al(2020) and Wang et al(2013).

9.What is the purpose of performing phylogenetic trees with mt genomes? These discoveries in your study cannot increase the clarity of Blattodea phylogeny.

Response:

In phylogenetic analyses, the absence of some species may lead to uncertain inferences about phylogenetic relationships (Tomita et al., 2011). Increased samplings can provide more complete and reliable information to perform phylogenetic analysis. In this study, we amplified the entire mt-genome sequence of Episymploce cockroaches for the first time. Performing phylogenetic trees with mt-genomes was used to confirm the taxonomic status of E.splendens.

10.Page 5, line 100. Authors should write the specific operation process.

Response:

Thanks for your suggestion. We rechecked the part of operation process and more detailed information were added. We supplemented “The concentration and purity of gDNA were detected by spectrophotometer. Firstly, 2 μl DNA elution solution was taken to calibrate the instrument. Then, the spot sample hole was wiped clean and 2 μl DNA was taken to determine the concentration and purity of the test sample. In addition, DNA was detected by agarose gel electrophoresis, and 1% agarose gel electrophoresis judged whether DNA was successfully extracted or not. Finally, DNA was stored at -20 ℃” here.

11.Page 10, line 183. Your purpose? The dataset you used here is not clear. 12S,16S,or PCG?

Response:

Thank you for your valuable advice. We amplified 2 Numts during PCR in this study. Divergence dating analysis was used to confirm the time of mitochondrial DNA transfer to nuclear DNA. The datasets we used here were aligned sequences, Numt-1 corresponded to partial lrRNA, and Numt-2 was similar to partial lrRNA and its neighboring tRNA-Val.

12.Page 10, line 184. Your aligned sequences used in Beast is not clear. You should demonstrate here instead of in the results part. 

Response:

Thanks for your constructive suggestion. We moved "There were two Numts found in E. splendens, namely Numt-1 and Numt-2. Through sequence alignment by MEGA 5.2, Numt-1 corresponded to partial lrRNA, and Numt-2 was similar to partial lrRNA and its neighboring tRNA-Val of the E. splendens mito-genome" from the results part to the methods part.

13.Page 11, line 101. Please select the right fossils according to Evargelista et al. (2017) and Evangelista et al (2019)

Response:

Thank you for your careful work. We read the references carefully and selected new fossils to calculate the divergence time, please see Table 2.

14.Page 9, line 154. This part should be ahead of divergence time analyses.

Response:

Thank you for your careful reading of our manuscript. We took the part “Phylogenetic inference” in front of “Divergence dating analysis”.

15.How long are your datasets? And why not use 12S and 16S?

Response:

Thank you for your valuable advice. The remaining length of nucleotide data of 13PCGs was 9,923bp, amino acid sequences of 13 PCGs was 10,032bp, and lrRNA+srRNA was 1,375bp. Besides, we added ML and BI trees based on 12S+16S.

16.There's no need to pick up so much termite data. You should choose more data of Ectobiidae instead of the data out of Blattodea.

Response:

Thank you for your suggestion. We selected 13 PCGs to performe phylogenetic analysis, but the number of complete mt genomes in Blattodea (expect for termites) especially in Ectobiidae were very limited. We took as many whole mt genomes as we could find in NCBI before, so we did not change the taxas used in this study.

17.Page 9, line 168. It’s not clear that the two datasets correspond to ML and BI models respectively.

Response:

Thank you for your careful work. For the 3 datasets, the GTR+I+G model was chosen as the best-fitting model for BI analysis, the GTR model was chosen as the best-fitting model for the ML analysis. We had revised phylogenetic inference in the methods part and made it more clear and organized. 

18.Page 12, line 214. Did you consider the sequencing error? I suggest another way to confirmed the gene spacer.

Response:

Thank you for your suggestion. Actually, we designed other primers for confirmation and got the same result (Primers in the table below). Besides, similar situation had been found in Blaptica dubia (Cheng et al, 2016). So we thought our judgment was correct.

 F R

1 TATGCTACCTTTGCACGGT TTATATCCTCCAGTAGATCCTACA

2 CTACTACCGTATGTTACGACTTA TGATTATATCCTCCAGTAGATCC

19.Page 12, line 220. Still need to be confirmed.

Response:

Thank you for your suggestion. For this problem, we also designed different primers to confirm our speculation (The primers were listed in the table below). Different primers amplified the same result. Besides, the deletion of tRNAs had been found in other animal mt genomes (San et al., 2004; Peng et al., 2006) before. So we believed our conclusion.

 F R

1 AATAAAGAGTGCCCCTGTC GATATTACAGGGATGAAGGATA

2 TAAAGAGTGCCCCTGTC TGATGAAGCTAAGGCTTGAA

20.Move to discussion.

Response:

Thank you for your suggestion. We deleted “However, the reason for the high content of AT in the mitochondrial genome is still unclear”.

21.Gene names need to be in italics and unified in this paper. For example, both ND1 and NAD1 appeared in this paper.

Response:

Thank you for your valuable advice. We unified the gene names and changed the gene names to italics in our manuscript.

22.Delete. Most of the content of your statement in this part does not belong to the results.

Response:

Thank you for your careful reading of our manuscript. We remove “The A+T rich region plays an important role in mitochondrial transcription and replication [47] [48]. Since the control region does not code for genes, the degree of variation in the control region is high and the length varies greatly due to its non-coding characteristic [49]. For example, the non-coding region in insects varies from 70 bp in Ruspolia dubia to 9-13 kb in Pissodes spp. [50] [51]” “Repeated units in the control regions were also found in different blattarian insects. However, these repeated sequences had low similarity in various cockroaches and thus may have different origins [4]”.

23. In this part, there is no new discovery based on your data. Although there are some difference with the previous study , I think it is because the different datasets were used.

Response:

Thank you for your suggestion. Phylogenetic analysis was used to confirm the taxonomic status of E.splendens.

24.Page 16, line 281. Write directly. Foe example, -value (BSP=@@, Fig. 4).

Response:

Thank you for your suggestion. We standardized writing in the manuscript, please see line 281, line 282 and line 284.

25.The name is inappropriate and doesn’t reflect the divergence time analyses.

Response:

Thank you for your suggestion. We changed “Numts” to “Numts and its divergence time”.

26.Still, it did not belong to the Results. References 54-56 are very old, please cite the lateset one.

Response:

Thank you for your suggestion. We deleted “Numts have been found in the nuclear genomes of various eukaryotes and can be used as outgroups of phylogeny to solve phylogenetic problems [54] [55] [56]. The two Numts found in this study could be useful tools for the study of cockroach phylogenetic relationships in the future”.

27.This is already a fact in many previous studies (2007-2021).

Response:

Thank you for your suggestion. We deleted “The Blattodea, Isoptera and Mantodea all belonged to Dictyoptera based on morphological and anatomical characteristics [11] [12]. According to molecular and morphology data, Mantodea was considered as the basal split within Dictyoptera [81] [82]. However, DeSalle et al [83] supported that Mantodea and Isoptera were sister groups within Dictyoptera. Nevertheless, more recent studies discovered that Mantodea was the sister clade to (Isoptera + Blattodea), and even indicated that Isoptera should be classified as a family (Termitidae) within Blattodea [34] [39] [84]. In our study, all results supported Mantodea was a sister clade to Blattodea (including Blattodea and Isoptera) with high support. In addition, we found Isoptera clustered into a monophyletic clade with high support (Fig 4 and S3 Fig), and this was also supported by several previous studies [9] [12] [85]. Consequently, we propose that Isoptera should be classified as a family within Blattodea”.

28.Page 23, line 446. Reference is not correct. Djernæs, 2018.

Response:

Thanks for your suggestion and sorry for the careless mistake. We had changed the right reference in the manuscript.

29.Page 23, line 447. Blattodea can be divided into three superfamilies: Blaberidae, Blattoidea and Corydioidea.

Response:

Thank you for your suggestion. We added "Blattodea can be divided into three superfamilies: Blaberidae, Blattoidea and Corydioidea" here.

30.Page 24. The author should refer to the latest literature before revising this part.

Response:

Thank you for your suggestion. We revised the phylogenetic analysis part and listed the latest literatures.

31.Page 24,25. When it comes to phylogenetic relationships, the author ignores the recent literatures. such as Wang et al., 2017; Bourguignon et al., 2018; Evangelista et al., 2019; Djernæs, et al.,2020 and so on.

Please rewrite it.

Response:

Thanks for your advice We modified the references and cited the recent literatures.

---

## [Decision Letter · Decision Letter 1]

11 Oct 2021

PONE-D-21-15397R1Complete mitochondrial genome of Episymploce splendens (Blattodea: Ectobiidae): a large intergenic spacer and lacking of two tRNA genesPLOS ONE

Dear Dr. Zhang,

Thank you for submitting your manuscript to PLOS ONE. After careful consideration, we feel that it has merit but does not fully meet PLOS ONE’s publication criteria as it currently stands. Therefore, we invite you to submit a revised version of the manuscript that addresses the points raised during the review process.

We look forward to receiving your revised manuscript.

Kind regards,

Tzen-Yuh Chiang

Academic Editor

PLOS ONE

Journal Requirements:

Additional Editor Comments (if provided):

Reviewers' comments:

Reviewer's Responses to Questions

**Comments to the Author**

1. If the authors have adequately addressed your comments raised in a previous round of review and you feel that this manuscript is now acceptable for publication, you may indicate that here to bypass the “Comments to the Author” section, enter your conflict of interest statement in the “Confidential to Editor” section, and submit your "Accept" recommendation.

Reviewer #1: (No Response)

Reviewer #2: All comments have been addressed

2. Is the manuscript technically sound, and do the data support the conclusions?

Reviewer #1: Partly

Reviewer #2: Yes

3. Has the statistical analysis been performed appropriately and rigorously? 

Reviewer #1: No

Reviewer #2: Yes

4. Have the authors made all data underlying the findings in their manuscript fully available?

Reviewer #1: Yes

Reviewer #2: Yes

5. Is the manuscript presented in an intelligible fashion and written in standard English?

Reviewer #1: Yes

Reviewer #2: Yes

6. Review Comments to the Author

Reviewer #1: Dear authors,

Thank you for your revised manuscript and your rebuttal letter. I appreciate all the work done to address the different concerns that were raised. However, I think that one main point has been ignored or misunderstood, and I really think it is a critical point. The point deals with your phylogenetic analyses. I focused my whole review on this single point because it affects deeply the whole structure of the manuscript. You will find a few additional details in the attached document but, again, my main point can be illustrated with those excerpts.

- l. 24-26: " The complete mitochondrial genome of Episymploce splendens, 15,802 bp in length, was determined and used to reconstruct the phylogenetic relationships within Blattodea in this study”.

There are plenty of recent phylogenetic studies for this group, with larger taxonomic and character sampling. Adding one species will not be significant enough to justify publication. And starting your abstract with this aim is not appropriate. Similarly, the end of your abstract deals with phylogenetic results, which cannot be a main output of your study. Your work is valuable but not for this.

- l. 67-72: “The phylogenetic results with different datasets and inference methods mays show different topologies, and the lack of some species in Blattodea may lead to some deviations. For example, the phylogenetic relationships between three superfamilies in Blattodea remain to be discussed further. Besides, the placement of Corydiidae is still in dispute”.

Even if one totally agrees with this summary of Blattodea phylogenetics, it is irrelevant as to what your study brings. Data for one Ectobiid species (whose genus as already been included in previous phylogenetic analyses) will not bring evidence as to the relationships between the three superfamilies or the placement of Corydiidae. This shows that the additional value of your work is not about cockroach phylogenetics. So please reframe your work, emphasizing its strength and not issues that cannot be tackled with your data.

- l. 82-83: “This study confirms the taxonomic status of E. splendens”. In your rebuttal letter “Performing phylogenetic trees with mt-genomes was used to confirm the taxonomic status of E. splendens”.

I think this reveals that the purpose of the phylogenetic analyses is unclear and I argue that even the second aim (i.e. confirming the taxonomic status) is inappropriate. I believe the taxonomic status of E. splendens is not controversial. So in a way, there is no need to confirm this status. In addition, performing a large scale phylogenetic analysis with only three Ectobiid species (the two others belonging to another genus, Blattella) cannot allow you to confirm this status. Finally, if I am wrong, then please provide the readers with the necessary elements to understand how controversial its taxonomic status was.

In relation to that, I want to underline one part of your rebuttal letter. You wrote “reviewer 2 suggested us to use more phylogenetic methods”. I have carefully read their comments but I did not find this. Instead, I understood that reviewer 2 as well found several issues in the phylogenetic study and I listed, for instance, “The authors should highlight the two NUMTS not the phylogenetic relationships within Blattodea” or “What is the purpose of performing phylogenetic trees with mt genomes? These discoveries in your study cannot increase the clarity of Blattodea phylogeny” or “In this part, there is no new discovery based on your data”. So using reviewer 2 comments to justify keeping the phylogenetic analyses is at odds with what I understood from their review and did not convince me that it is a major issue with your work, which again is valuable, but not in this regard.

Best wishes,

Reviewer #2: The author has done a thorough work to solve the problems I listed in the initial round of review. I am satisfied with the amendments and clarifications made to my concerns at this time. It seems that they have also solved many of the problems of reviewer 1. After this round, the manuscript has been greatly improved.

There are still some errors in the manuscript,ig.Crypyocercidae, please check all scientific names.

7. PLOS authors have the option to publish the peer review history of their article (what does this mean?). If published, this will include your full peer review and any attached files.

Reviewer #1: No

Reviewer #2: No

---

## [Author Response · Author response to Decision Letter 1]

23 Oct 2021

Reviewer #1:

Thank you for your revised manuscript and your rebuttal letter. I appreciate all the work done to address the different concerns that were raised. However, I think that one main point has been ignored or misunderstood, and I really think it is a critical point. The point deals with your phylogenetic analyses. I focused my whole review on this single point because it affects deeply the whole structure of the manuscript. You will find a few additional details in the attached document but, again, my main point can be illustrated with those excerpts.

Response:

We really appreciate your valuable time on reviewing our manuscript. We disscussed your comments, and we thought they were useful and valuable. We did not get any new results from the phylogenetic analysis. The part phylogenetic analysis seems needless. So we deleted this part in our manuscript.The following are point by point responses to your comments:

l. 24-26: “The complete mitochondrial genome of Episymploce splendens, 15,802 bp in length, was determined and used to reconstruct the phylogenetic relationships within Blattodea in this study”.

There are plenty of recent phylogenetic studies for this group, with larger taxonomic and character sampling. Adding one species will not be significant enough to justify publication. And starting your abstract with this aim is not appropriate. Similarly, the end of your abstract deals with phylogenetic results, which cannot be a main output of your study. Your work is valuable but not for this.

Response:

Thanks for your suggestion. We changed “The complete mitochondrial genome of Episymploce splendens, 15,802 bp in length, was determined and used to reconstruct the phylogenetic relationships within Blattodea in this study” to “The complete mitochondrial genome of Episymploce splendens, 15,802 bp in length, was determined and annotated in this study”.

2. 67-72: “The phylogenetic results with different datasets and inference methods mays show different topologies, and the lack of some species in Blattodea may lead to some deviations. For example, the phylogenetic relationships between three superfamilies in Blattodea remain to be discussed further. Besides, the placement of Corydiidae is still in dispute”.

Even if one totally agrees with this summary of Blattodea phylogenetics, it is irrelevant as to what your study brings. Data for one Ectobiid species (whose genus as already been included in previous phylogenetic analyses) will not bring evidence as to the relationships between the three superfamilies or the placement of Corydiidae. This shows that the additional value of your work is not about cockroach phylogenetics. So please reframe your work, emphasizing its strength and not issues that cannot be tackled with your data.

Response:

Thanks for your advice. We discussed the comments of reviewers, and we considered these comments were valuable and correct. We did not get any new results from the phylogenetic analysis, and one species will not bring evidence as to the relationships between the three superfamilies or the placement of Corydiidae. So we deleted phylogenetic analysis in our manuscript.

3. 82-83: “This study confirms the taxonomic status of E. splendens”. In your rebuttal letter “Performing phylogenetic trees with mt-genomes was used to confirm the taxonomic status of E. splendens”.

I think this reveals that the purpose of the phylogenetic analyses is unclear and I argue that even the second aim (i.e. confirming the taxonomic status) is inappropriate. I believe the taxonomic status of E. splendens is not controversial. So in a way, there is no need to confirm this status. In addition, performing a large scale phylogenetic analysis with only three Ectobiid species (the two others belonging to another genus, Blattella) cannot allow you to confirm this status. Finally, if I am wrong, then please provide the readers with the necessary elements to understand how controversial its taxonomic status was.

Response:

Thanks for your suggestion. The taxonomic status of E. splendens is not controversial, and we we should delete phylogenetic analysis in our manuscript.

In relation to that, I want to underline one part of your rebuttal letter. You wrote “reviewer 2 suggested us to use more phylogenetic methods”. I have carefully read their comments but I did not find this. Instead, I understood that reviewer 2 as well found several issues in the phylogenetic study and I listed, for instance, “The authors should highlight the two NUMTS not the phylogenetic relationships within Blattodea” or “What is the purpose of performing phylogenetic trees with mt genomes? These discoveries in your study cannot increase the clarity of Blattodea phylogeny” or “In this part, there is no new discovery based on your data”. So using reviewer 2 comments to justify keeping the phylogenetic analyses is at odds with what I understood from their review and did not convince me that it is a major issue with your work, which again is valuable, but not in this regard.

Response:

Thanks for your careful work. Our discoveries in this study cannot increase the clarity of Blattodea phylogeny, so we deleted the phylogenetic analysis part.

Reviewer #2:

The author has done a thorough work to solve the problems I listed in the initial round of review. I am satisfied with the amendments and clarifications made to my concerns at this time. It seems that they have also solved many of the problems of reviewer 1. After this round, the manuscript has been greatly improved.

There are still some errors in the manuscript, ig.Crypyocercidae, please check all scientific names.

Response:

Thanks for your careful work. We have checked the contents of our manuscript.

---

## [Decision Letter · Decision Letter 2]

7 Dec 2021

PONE-D-21-15397R2Complete mitochondrial genome of  Episymploce splendens  (Blattodea: Ectobiidae): a large intergenic spacer and lacking of two tRNA genesPLOS ONE

Dear Dr. Zhang,

Thank you for submitting your manuscript to PLOS ONE. After careful consideration, we feel that it has merit but does not fully meet PLOS ONE’s publication criteria as it currently stands. Therefore, we invite you to submit a revised version of the manuscript that addresses the points raised during the review process.

We look forward to receiving your revised manuscript.

Kind regards,

Tzen-Yuh Chiang

Academic Editor

PLOS ONE

Reviewers' comments:

Reviewer's Responses to Questions

**Comments to the Author**

1. If the authors have adequately addressed your comments raised in a previous round of review and you feel that this manuscript is now acceptable for publication, you may indicate that here to bypass the “Comments to the Author” section, enter your conflict of interest statement in the “Confidential to Editor” section, and submit your "Accept" recommendation.

Reviewer #2: All comments have been addressed

Reviewer #3: (No Response)

2. Is the manuscript technically sound, and do the data support the conclusions?

Reviewer #2: Yes

Reviewer #3: No

3. Has the statistical analysis been performed appropriately and rigorously? 

Reviewer #2: Yes

Reviewer #3: N/A

4. Have the authors made all data underlying the findings in their manuscript fully available?

Reviewer #2: Yes

Reviewer #3: No

5. Is the manuscript presented in an intelligible fashion and written in standard English?

Reviewer #2: Yes

Reviewer #3: No

6. Review Comments to the Author

Reviewer #2: This round you responded all the comments carefully, and deleted the meaningless or not closely related part. it feels much more compact and focused. I suggest emphasizing the existence of pseudogenes(numts) and the time when pseudogenes appear in this cockroach, instead of a large intergenic spacer. And I think the introduction to the study of the cockroach mitochondrial genome is too simplistic.

Reviewer #3: I failed to remember that I have reviewed this paper before. I failed to find this mt genome of OK094023 in NCBI. I need this sequence to judge the mt genome right or not! If I reviewed it before, I must be concerned on this question. I did not believe two trnQ and trnM lost in Episymploce splendens.

The two trnQ fand trnM can be translated into between two CRs. As the authors said that this is the first report of the tRNAs deletion in Orthopteroidea mito-genomes. But I did not believe it. I suggest the authors check it carefully using the other species in genus Episymploce. Because if the mt genome is existed two similar CRs, the lost genes will be found between two CRs.

I suggest the authors should be carefully checked again.

When the authors used a pair of primers of "Es11 13822-CAGATTATATTGATTCGCACAAC" and "303-ATAGAACTGATGAAGCTAAGGC" to amplify PCR using 75 second Extension. It may be failed to amplify two control regions. I suggest the authors can design more primers to check it.

Two Numts were found by PCR. Two Numts were similar to lrRNA, which can be explained why the author failed to obtained trnQ and trnM. Check the mt genome again!

Another question is how the authors judge the two Numts ？ why is not lrRNA belonging to Numt？

I did think the divergence time of Numts are interesting. Because the title is "Complete mitochondrial genome of Episymploce splendens (Blattodea: Ectobiidae): a large intergenic spacer and lacking of two tRNA genes" not including Numts.

Is a large intergenic spacer similar to trnI or trnQ? or AT-rich region?

I read the paper and found the English writing is really poor. There are many mistaken in the paper. I can not believe the paper is revsied the second version.

All genes name should be the same in text, figures, and tables.

7. PLOS authors have the option to publish the peer review history of their article (what does this mean?). If published, this will include your full peer review and any attached files.

Reviewer #2: No

Reviewer #3: No

---

## [Author Response · Author response to Decision Letter 2]

16 Mar 2022

Response to Reviewers

Dear editor:

Thank you for giving us the opportunity to revise our manuscript entitled “Complete mitochondrial genome of Episymploce splendens (Blattodea: Ectobiidae): a large intergenic spacer and lacking of two tRNA genes” to PLOS ONE. We appreciate that the reviewers′ valuable comments for our manuscript. These comments are very important for improving our manuscript. We have revised our manuscript according to the reviewers′ comments and also made other improvement for our manuscript. We do hope that our manuscript is appropriate and could be considered for publication in this journal. 

This study was funded by the Special funds for central government to guide local scientific and Technological Development (2020ZYD098) and National Natural Science Foundation of China (U21A20409).The funders had no role in study design, data collection and analysis, decision to publish, or preparation of the manuscript.

Best regards,

Sincerely yours,

Xiuyue Zhang, PhD and Professor

College of Life Sciences, Sichuan University

Email: zhangxy317@126.com

Reviewers' comments:

Reviewer 2:

This round you responded all the comments carefully, and deleted the meaningless or not closely related part. it feels much more compact and focused. I suggest emphasizing the existence of pseudogenes (numts) and the time when pseudogenes appear in this cockroach, instead of a large intergenic spacer. And I think the introduction to the study of the cockroach mitochondrial genome is too simplistic.

Response:

Thanks for your suggestion. The mito-genome commonly displays exceptional economy of organization and the large intergenic spacer was uncommon. We believed the find of large intergenic spacer in mitochondrial genome of Episymploce splendens was important for the study of insect mitochondrial genome evolution in the future and so we retained it. And based on your suggests, we also further emphasized the discovery of numts and their significance, and added a description of the mitochondrial genome and enriched the content.

Reviewer 3:

1.I failed to remember that I have reviewed this paper before. I failed to find this mt genome of OK094023 in NCBI. I need this sequence to judge the mt genome right or not! If I reviewed it before, I must be concerned on this question. I did not believe two trnQ and trnM lost in Episymploce splendens.

The two trnQ f and trnM can be translated into between two CRs. As the authors said that this is the first report of the tRNAs deletion in Orthopteroidea mito-genomes. But I did not believe it. I suggest the authors check it carefully using the other species in genus Episymploce. Because if the mt genome is existed two similar CRs, the lost genes will be found between two CRs.

I suggest the authors should be carefully checked again.

When the authors used a pair of primers of "Es11 13822-CAGATTATATTGATTCGCACAAC" and "303-ATAGAACTGATGAAGCTAAGGC" to amplify PCR using 75 second Extension. It may be failed to amplify two control regions. I suggest the authors can design more primers to check it.

Response:

Thanks for your careful reading.

We really appreciate your valuable time on reviewing our manuscript. We had provided the complete mt-genome to Genbank and obtained the GenBank identifier (OK094023) but we set up the release date, and so this sequence cannot be found in NCBI at present. However, in order to the convenience of review, it has been uploaded as an attachment in our submitted manuscript and also in our revised manuscript.

The non-coding region of E. splendens mito-genome was located between srRNA and tRNA-Ile, and it was 290 bp long consist of two 120 bp long repeated units. The two lost tRNAs were located behind tRNA-Ile, and the interval between tRNA-Ile and NAD2 was only 24bp. They cannot form the 2 tRNAs. Because we used ARWEN and tRNA-SE to identify tRNAs and obtained same result through reference to secondary structure models for those genes from other blattarian insects (Ma et al., 2017; Xiao et al., 2012). The tRNA-Gln and tRNA-Met located between tRNA-Ile and NAD2, and they were typically 150bp long in other cockroaches. However, this site was only 24bp long in E. splendens. In addition, we tried to refer to other cockroach mitochondrial genomes, but however, there was no mitochondrial sequences reported in Episymploce in NCBI. In reported cockroach mitochondrial genomes, we did not find two similar CRs, namely, these reported cockroach mitochondrial genomes have only one A-T region. We designed different primers (in the table) to amplify this area and obtain same amplification sequences, so we think our experiment result was correct and there was no sequence lost due to inappropriate primers. We could not found the secondary structures of two tRNAs in E. splendens mito-genome sequence, and so we believed their loss event. Although tRNA lacking was rare, other species had been found in previous studies. For example, Limnonectes bannaensis mt-genome lost 4 tRNAs (tRNA-Ala, tRNA-Asn, tRNA-Cys and tRNA-Glu), Gegeneophis ramaswamii mt-genome lost tRNA-Phe, Urochela quadrinotata Reuter mt-genome lost 2 tRNAs (tRNA-Ile and tRNA-Gln), Platysternon megacephalum mt-genome lost tRNA-Thr. These loss events of tRNAs occur accidentally and independently in these mt-genome, and we also believe that these findings will be useful for future studies on the cause of their loss and the subsequent adaptive evolution.

 F R

1.ES11 CAGATTATATTGATTCGCACAAC ATAGAACTGATGAAGCTAAGGC

2.(Ma et al,2017) CCTCTAAAAAGACTAAAATACCGCC GGAATCATCAGTGAAAGGGAGC

3. AATAAAGAGTGCCCCTGTC GATATTACAGGGATGAAGGATA

4. TAAAGAGTGCCCCTGTC TGATGAAGCTAAGGCTTGAA

2. Two Numts were found by PCR. Two Numts were similar to lrRNA, which can be explained why the author failed to obtained trnQ and trnM. Check the mt genome again!

Another question is how the authors judge the two Numts？ Why is not lrRNA belonging to Numt？

I did think the divergence time of Numts are interesting. Because the title is "Complete mitochondrial genome of Episymploce splendens (Blattodea: Ectobiidae): a large intergenic spacer and lacking of two tRNA genes" not including Numts.

Response:

The sequences which originated from the invasion of mtDNA into nuclear DNA were named Numts. Numts exist in the nuclear DNA and do not belong to the mitochondrial genome sequence, but due to their partial sequences may be identical to their source sequences, some primers that amplify mitochondrial genomes may accidentally amplify these sequences. So, two Numts may be not related to the loss of tRNAs.

 Primers that amplified mitochondrial genes may occasionally amplify Numts. In this study, agarose gel electrophoresis of the amplification product (the primer Numt-1-1 and Numt-2-1, Table 1) appeared as two bands. We then redesigned primers (Numt-1-2 and Numt-2-2, Table 1) to amplify these regions and obtained the same result. These non-targeted bands were purified and sequenced, and obtained sequences belonging to mitochondria via the analysis of NCBI blast. Due to differences in mutation rates or relaxation from selection pressure, they had some degree of base differences compared to corresponding mito-genome sequences. These sequences have more mutations of insertion and deletion compared to their corresponding mitochondrial sequences, and may lose their functions. They did not belong to the mitochondrial genome, so they were not the lost tRNAs. We had explained this in the corresponding section of the manuscript (Page line 162). Numts can be used to solve problems in phylogeny, but they were not a part of the mitochondria, so we did not mentioned them in the title. 

3.Is a large intergenic spacer similar to trnI or trnQ? or AT-rich region?

Response:

The large intergenic spacer was not A-T region or similar to trnI or trnQ. It was similar to its corresponding srRNA with a 64.6% similarity.

4.I read the paper and found the English writing is really poor. There are many mistaken in the paper. I can not believe the paper is revsied the second version.

Response:

we're sorry for the writing mistaken and we have made careful revisions to our English writing.

5.All genes name should be the same in text, figures, and tables.

Response:

We have made relevant changes in our revised manuscript.

---

## [Decision Letter · Decision Letter 3]

11 Apr 2022

PONE-D-21-15397R3Complete mitochondrial genome of  Episymploce splendens  (Blattodea: Ectobiidae): a large intergenic spacer and lacking of two tRNA genesPLOS ONE

Dear Dr. Zhang,

Thank you for submitting your manuscript to PLOS ONE. After careful consideration, we feel that it has merit but does not fully meet PLOS ONE’s publication criteria as it currently stands. Therefore, we invite you to submit a revised version of the manuscript that addresses the points raised during the review process.

We look forward to receiving your revised manuscript.

Kind regards,

Tzen-Yuh Chiang

Academic Editor

PLOS ONE

Reviewers' comments:

Reviewer's Responses to Questions

**Comments to the Author**

1. If the authors have adequately addressed your comments raised in a previous round of review and you feel that this manuscript is now acceptable for publication, you may indicate that here to bypass the “Comments to the Author” section, enter your conflict of interest statement in the “Confidential to Editor” section, and submit your "Accept" recommendation.

Reviewer #3: All comments have been addressed

Reviewer #4: (No Response)

2. Is the manuscript technically sound, and do the data support the conclusions?

Reviewer #3: Yes

Reviewer #4: Partly

3. Has the statistical analysis been performed appropriately and rigorously? 

Reviewer #3: Yes

Reviewer #4: No

4. Have the authors made all data underlying the findings in their manuscript fully available?

Reviewer #3: Yes

Reviewer #4: No

5. Is the manuscript presented in an intelligible fashion and written in standard English?

Reviewer #3: Yes

Reviewer #4: No

6. Review Comments to the Author

Reviewer #3: (No Response)

Reviewer #4: This manuscript described the molecular features of the mtDNA of the blattarian species Episymploce splendens, with major emphasis to the presumed lack of two tRNA encoding genes, the occurrence of intergenic spacers and on the occurrence of presumed Numts.

Central to the processed analyses is the impossibility to identify, along the mtDNA, genes encoding trnM and trnQ. It is quite difficult to understand if unsuccessful scanning for tRNA genes is due to the inability of automated programs to detect sequences of abnormal size and structure. Some arthropods species have highly modified tRNA secondary structures and their occurrence within a mtDNA may be nightmare to be assessed. In some instances, genes encoding for tRNAs are embedded within both AT-rich region and PCG encoding genes. I understand Authors have scanned manually for their occurrence and trust them when they reach to the conclusion that every attempt was unsuccessful. However, I suggest to further investigate the occurrence of the missing tRNAs within the coding sequence of adjoining genes (i.e. partially overlapping with tRNAI and with nad2). This would possibly be done by considering every possible trinucleotide primary sequence, corresponding to the anticodon of the missing tRNAs, in the vicinity of the site where these latter are usually found. One more additional analysis should be addressed using relative taxa (other species of the same genus). There is a chance that sequencing the same DNA fragment in closely related taxa would support the presumed lack of the tRNA genes, as well. I understand this would need further analyses but would also provide a more convincing support to the conclusions.

Another result of Authors’s study was the presumed discovery of mitochondrial sequences possibly inserted within the nuclear genome. I’m more prone to suspect that this may be due to some artifact of the sequencing procedure, although I don’t have access to supplementary files S3 and S4. I would rather write a more prudential conclusion: the occurrence of Numts is not clearly assessed, although suspected.

Other minor correction along the text.

In conclusion, the lack of tRNA encoding genes should be further investigated and the occurrence of Numts should be downscaled to a possibility, not stated evidence. These before the manuscript should be considered for publication.

7. PLOS authors have the option to publish the peer review history of their article (what does this mean?). If published, this will include your full peer review and any attached files.

Reviewer #3: No

Reviewer #4: No

---

## [Author Response · Author response to Decision Letter 3]

16 Apr 2022

Response to Reviewers

Dear editor:

Thank you for giving us the opportunity to revise our manuscript entitled “Complete mitochondrial genome of Episymploce splendens (Blattodea: Ectobiidae): a large intergenic spacer and lacking of two tRNA genes” to PLOS ONE. We appreciate that the reviewers′ valuable comments for our manuscript. These comments are very important for improving our manuscript. We have revised our manuscript according to the reviewers′ comments and also made other improvement for our manuscript. We do hope that our manuscript is appropriate and could be considered for publication in this journal. 

Best regards,

Sincerely yours,

Xiuyue Zhang, PhD and Professor

College of Life Sciences, Sichuan University

Email: zhangxy317@126.com

Reviewers' comments:

Reviewer 4:

1.This manuscript described the molecular features of the mtDNA of the blattarian species Episymploce splendens, with major emphasis to the presumed lack of two tRNA encoding genes, the occurrence of intergenic spacers and on the occurrence of presumed Numts.

Central to the processed analyses is the impossibility to identify, along the mtDNA, genes encoding trnM and trnQ. It is quite difficult to understand if unsuccessful scanning for tRNA genes is due to the inability of automated programs to detect sequences of abnormal size and structure. Some arthropods species have highly modified tRNA secondary structures and their occurrence within a mtDNA may be nightmare to be assessed. In some instances, genes encoding for tRNAs are embedded within both AT-rich region and PCG encoding genes. I understand Authors have scanned manually for their occurrence and trust them when they reach to the conclusion that every attempt was unsuccessful. However, I suggest to further investigate the occurrence of the missing tRNAs within the coding sequence of adjoining genes (i.e. partially overlapping with tRNAI and with nad2). This would possibly be done by considering every possible trinucleotide primary sequence, corresponding to the anticodon of the missing tRNAs, in the vicinity of the site where these latter are usually found. One more additional analysis should be addressed using relative taxa (other species of the same genus). There is a chance that sequencing the same DNA fragment in closely related taxa would support the presumed lack of the tRNA genes, as well. I understand this would need further analyses but would also provide a more convincing support to the conclusions.

Another result of Authors’s study was the presumed discovery of mitochondrial sequences possibly inserted within the nuclear genome. I’m more prone to suspect that this may be due to some artifact of the sequencing procedure, although I don’t have access to supplementary files S3 and S4. I would rather write a more prudential conclusion: the occurrence of Numts is not clearly assessed, although suspected.

Other minor correction along the text.

In conclusion, the lack of tRNA encoding genes should be further investigated and the occurrence of Numts should be downscaled to a possibility, not stated evidence. These before the manuscript should be considered for publication.

Response:

We really appreciate your valuable time on reviewing our manuscript. We have tried to investigate the occurrence of the missing tRNAs within the coding sequence of adjoining genes based on your good comments and don’t have found the missing tRNAs. 

The tRNA-Gln and tRNA-Met located between tRNA-Ile and NAD2, and they were typically 150bp long in other blattarian insects. However, the interval between tRNA-Ile and NAD2 in E. splendens mit-genome was only 24bp and cannot form the 2 tRNAs. We designed different primers to amplify this area and obtain same amplification sequences, so we think our experiment result was correct and there was no sequence lost due to inappropriate primers. Because we used ARWEN and tRNA-SE to identify tRNAs and obtained same result through reference to secondary structure models for those genes from other blattarian insects (Ma et al., 2017; Xiao et al., 2012). 

Based on your suggestion, we extended the 24bp sequence between tRNA-ILE and NAD2 about 70bp into tRNA-ILE and NAD2, respectively. Then use the online server ARWEN (http://mbio-serv2.mbioekol.lu.se/ARWEN/) and tRNA-SE to identify tRNAs, and we found no tRNAs (as the following Figure 1).

Figure 1 Result in ARWEN

Using similar sequence interception methods, we mapped RNA structures using RNAfold (an online tool for mapping RNA structures) with tRNA-Gln and tRNA-Met anticodons as the centre, and found no typical tRNAs. The partial results are shown in the following Figure 2.

Figure 2 Part of the result in RNAfold（http://rna.tbi.univie.ac.at//cgi-bin/RNAWebSuite/RNAfold.cgi）

We did also the similar analysis with AT-rich region and other PCG encoding genes, and didn’t find the absent tRNAs. So, we believed the loss event dose exist in E. splendens mito-genome.

Only Episymploce splendens mit-genome has been sequenced in the Episymploce (our study), so we can’t do additional analysis within Episymploce. It is a pity that we don't have samples of other species of this genus, so we can't sequence the same fragments from other species and can’t know if there are similar tRNAs loss in other species. Although related species may have similar events, some previous studies also showed that the loss of tRNAs occurs accidentally or independently in different species (Shi et al, 2021). Therefore, Further studies are needed in the future to confirm whether other species of Episymploce also have tRNAs loss. Anyway, 2 species of Blattellidae (EU854321 and JX233805) have been sequenced, and neither of them has any tRNA lossing. 

Indeed, sequencing, especially poor sequencing, can lead to artifacts of Numts, but our sequencing results were very good (As the sequencing peaks of Numts shown in following Figure 3, 4). We used forward and reverse bidirectional sequencing, and removed the inaccurate sequence at the beginning and end of sequences, and finally obtained the correct spliced sequences. And we also used different primer pairs to perform PCR and sequence, and we obtained the same results, so we believed that our results are correct. These Numts have more mutations of insertion and deletion compared to their corresponding mitochondrial sequences due to differences in mutation rates or relaxation from selection pressure. The supplementary files S3 and S4 showing the sequence alignment of Numts with their corresponding mtDNA were shown in following Figure 5,6.

Figure 3 The sequencing peaks of Numt-1

Figure 4 The sequencing peaks of Numt-2

Figure 5 Fig S3 Sequence alignment of Numt-1 and the corresponding mtDNA

Figure 6 Fig S4 Sequence alignment of Numt-2 and the corresponding mtDNA

2. Page 2, line 32. Here and along the text... blattarian not capital letter

Response:

Thanks for your suggestion and sorry for the careless mistake. We have checked the text and corrected the mistake capital letter.

3. Page 3, line 56-58. rewrite sentence for clarity

Response:

Thanks for your suggestion. We changed “The tRNAs are most various in no-typical insect mito-genomes” to “The changes of tRNAs are most various in no-typical insect mito-genomes”.

4. Page 3, line 58. obvious

Response:

Phthiraptera and Psocoptera insect mito-genomes may have the most changes, that they could fission into fragmented mito-genomes and have numerous pseudo-genes and diverse gene rearrangements (Simon et al, 2013; Chen et al, 2016; Shi et al, 2021; Shi et al, 2016).

5. Page 3, line 59. with respect to?

Response:

Thanks for your suggestion. Their fast evolution were shown in Page 3, line 61-62 “that they could fission into fragmented mito-genomes and have numerous pseudo-genes and diverse gene rearrangements”(Simon et al, 2013; Chen et al, 2016; Shi et al, 2021; Shi et al, 2016).

6. Page 3, line 61. s

Response:

Thanks for your suggestion and sorry for the careless mistake. We have changed “rearrangement” to “rearrangements”.

7. Page 3, line 66. what is a "conserved evolution"?

Response:

We has changed “The blattarian mito-genomes seem to be in a conserved evolution”. to “The blattarian mito-genomes seem to be conserved in evolution”.

8. Page 4, line 77. what are the open questions and aims addressed by Authors? Please add aims

Response:

Thanks for your suggestion. We have changed “Episymploce splendens belongs to Ectobiidae and at present, no complete mitochondrial sequences of Episymploce have been recorded in the NCBI database. Previous studies have mainly focused on the morphological features of Episymploce rather than molecular data [18-19]. In this study, we sequenced ” to “Episymploce splendens belongs to Ectobiidae and Previous studies have mainly focused on the morphological features of Episymploce rather than molecular data [18-19]. At present, no complete mitochondrial sequences of Episymploce have been recorded in the NCBI database. To obtain the sequence information and organization features of mito-genomes in Episymploce, we sequenced”.

9. Page 4, line 81-86. this is not M&M topic

Response:

Thank you for your valuable advice. We have deleted the irrelevant content.

10. Page 5, line 92-95. delete

Response:

Thank you for your suggestion. We have deleted this part.

11. Page 8, line117. "invasion" is a term used for transposons

Response:

Thank you for your suggestion. We have changed this sentence to “The sequences which were transferred to nuclear DNA from mitochondrial genome were named Numts”.

12. Page 8, line 137. checked

Response:

Thanks for your suggestion. We have changed “recognized” to “checked”.

13. Page 10, line 169. no genome codes for genes, but rather for proteins of other transcripts

Response:

Thanks for your advice. We have changed “It coded for 35 mitochondrial genes:” to “It contained 35 mitochondrial genes:”

14. Page 11, line 183. usually

Response:

Thanks for your advice. We have added “usually” after “tRNAs”.

15. Page 14, line 238. what is a "low rate of evolution"?

Response:

.We have changed this sentence to “They have different mutation rates compared to their ancient mtDNA”.

Thank you again for your careful revision and good suggestions for our manuscript.

---

## [Decision Letter · Decision Letter 4]

22 Apr 2022

Complete mitochondrial genome of  Episymploce splendens  (Blattodea: Ectobiidae): a large intergenic spacer and lacking of two tRNA genes

PONE-D-21-15397R4

Dear Dr. Zhang,

We’re pleased to inform you that your manuscript has been judged scientifically suitable for publication and will be formally accepted for publication once it meets all outstanding technical requirements.

Kind regards,

Tzen-Yuh Chiang

Academic Editor

PLOS ONE

Additional Editor Comments (optional):

Reviewers' comments:

Reviewer's Responses to Questions

**Comments to the Author**

1. If the authors have adequately addressed your comments raised in a previous round of review and you feel that this manuscript is now acceptable for publication, you may indicate that here to bypass the “Comments to the Author” section, enter your conflict of interest statement in the “Confidential to Editor” section, and submit your "Accept" recommendation.

Reviewer #3: All comments have been addressed

2. Is the manuscript technically sound, and do the data support the conclusions?

Reviewer #3: Yes

3. Has the statistical analysis been performed appropriately and rigorously? 

Reviewer #3: Yes

4. Have the authors made all data underlying the findings in their manuscript fully available?

Reviewer #3: Yes

5. Is the manuscript presented in an intelligible fashion and written in standard English?

Reviewer #3: Yes

6. Review Comments to the Author

Reviewer #3: I think the paper can be accepted now. Congratulation! All comments have be well responsed.

Although the gene loss is hard to find, the authors recheck all sequences. So I have no comment now.

7. PLOS authors have the option to publish the peer review history of their article (what does this mean?). If published, this will include your full peer review and any attached files.

Reviewer #3: No

---

## [Editor Report · Acceptance letter]

23 May 2022

PONE-D-21-15397R4 

Complete mitochondrial genome of *Episymploce splendens* (Blattodea: Ectobiidae): a large intergenic spacer and lacking of two tRNA genes 

Dear Dr. Zhang:

I'm pleased to inform you that your manuscript has been deemed suitable for publication in PLOS ONE. Congratulations! Your manuscript is now with our production department. 

Kind regards, 

on behalf of

Dr. Tzen-Yuh Chiang 

Academic Editor

PLOS ONE